# Precursor region with full phonon softening above the charge-density-wave phase transition in 2*H*-TaSe$_2$

Xingchen Shen [1,2], Rolf Heid [1], Roland Hott[1], Amir-Abbas Haghighirad [1], Björn Salzmann [3], Marli dos Reis Cantarino [3,4], Claude Monney [3], Ayman H. Said[5], Mehdi Frachet[1], Bridget Murphy [6,7], Kai Rossnagel [6,7,8], Stephan Rosenkranz [9] & Frank Weber [1] ✉

Research on charge-density-wave (CDW) ordered transition-metal dichalcogenides continues to unravel new states of quantum matter correlated to the intertwined lattice and electronic degrees of freedom. Here, we report an inelastic x-ray scattering investigation of the lattice dynamics of the canonical CDW compound 2*H*-TaSe$_2$ complemented by angle-resolved photoemission spectroscopy and density functional perturbation theory. Our results rule out the formation of a *central-peak* without full phonon softening for the CDW transition in 2*H*-TaSe$_2$ and provide evidence for a novel precursor region above the CDW transition temperature $T_{CDW}$, which is characterized by an overdamped phonon mode and not detectable in our photoemission experiments. Thus, 2*H*-TaSe$_2$ exhibits structural before electronic static order and emphasizes the important lattice contribution to CDW transitions. Our ab-initio calculations explain the interplay of electron-phonon coupling and Fermi surface topology triggering the CDW phase transition and predict that the CDW soft phonon mode promotes emergent superconductivity near the pressure-driven CDW quantum critical point.

Charge-density-wave (CDW) order, a periodic modulation of the charge carrier density, often coexists or competes with superconductivity, e.g., in copper-oxide superconductors[1-4] and transition-metal-based kagome metals[5-10]. Strong electronic correlations originating in the electronic band structures featuring flat bands, Dirac cones and van Hove singularities are expected to be responsible for the reported phase competition. On the other hand, classic CDW materials such as the layered transition-metal dichalcogenides (TMD) continue to provide a rich playground for emergent physics including excitonic insulators[11-13], dimensionality-dependent correlated electronic phases[14,15] as well as intriguing magnetic properties[16] with potential applications in spintronic devices[17]. CDW order is widespread in metallic TMDs[18,19]. In the seminal model by Peierls[20] an electronic instability in the presence of finite electron–phonon coupling (EPC) stabilizes the CDW ground state. While this scenario applies to several known quasi-1D materials[21-24], recent work for layered TMDs[25-28] showed that the full momentum dependence of both the electronic band structure and the EPC have to be taken into account to explain CDW formation and the existence of closely related superconducting phases[29-34].

[1]Institute for Quantum Materials and Technologies, Karlsruhe Institute of Technology, 76021 Karlsruhe, Germany. [2]College of Physics, Chongqing University, Chongqing 401331, P. R. China. [3]Département de Physique and Fribourg Center for Nanomaterials, Université de Fribourg, 1700 Fribourg, Switzerland. [4]Instituto de Física, Universidade de São Paulo, São Paulo, São Paulo 05508-090, Brazil. [5]Advanced Photon Source, Argonne National Laboratory, Lemont, IL 60439, USA. [6]Institute of Experimental and Applied Physics and KiNSIS, Kiel University, 24098 Kiel, Germany. [7]Ruprecht Haensel Laboratory, Kiel University, 24098 Kiel, Germany. [8]Ruprecht Haensel Laboratory, Deutsches Elektronen-Synchrotron DESY, 22607 Hamburg, Germany. [9]Materials Science Division, Argonne National Laboratory, Lemont, IL 60439, USA. ✉e-mail: frank.weber@kit.edu

A hallmark of CDW order is a periodic lattice distortion and the softening of a symmetry-related phonon to zero energy at the phase transition temperature $T_{CDW}$. Thus, investigation of this soft phonon mode can yield important insights about the underlying mechanism governing the phase transition and has been indispensable to understanding the physics of many CDW materials[24,25,27,28]. However, a full softening is not always observed questioning the displacive nature of the CDW phase transition. For instance, quasi-1D materials NbSe$_3$[35] and (TaSe$_4$)$_2$I[36] as well as cuprates[2,3,37] and kagome metals[8] feature no or only little softening at $T_{CDW}$.

2$H$-TaSe$_2$ is a prototypical CDW compound featuring a large periodic lattice distortion[38] and a momentum-dependent energy gap in the electronic band structure in its low-temperature state[39,40]. It is a layered material (see inset in Fig. 1a) for which CDW order with a transition temperature $T_{CDW} = 122.3$ K was reported in the 1970s[41]. On cooling through $T_{CDW}$, 2$H$-TaSe$_2$ first enters a CDW phase with an incommensurate ordering wave vector $\mathbf{q}_{CDW} = (0.323,0,0)$ (all wave vectors are given in reciprocal lattice units; see the "Methods" section), which gradually evolves on cooling and reaches the commensurate value $\mathbf{q}_{CDW} = (1/3,0,0)$ at $T_{CDW-C} \approx 90$ K[42]. The superconducting transition temperature of 2$H$-TaSe$_2$ is $T_{sc} = 0.1$ K. However, $T_{sc}$ increases up to 8.2 K under pressure and a close connection between fluctuations of the pressure-suppressed CDW order and superconductivity is anticipated[31]. In 2$H$-TaSe$_2$, the CDW soft phonon mode corresponds to the longitudinal acoustic (LA) phonon propagating along the [100] direction and has been investigated by inelastic neutron scattering (INS) experiments in the 1970s[38,41]. Surprisingly, the observed phonon

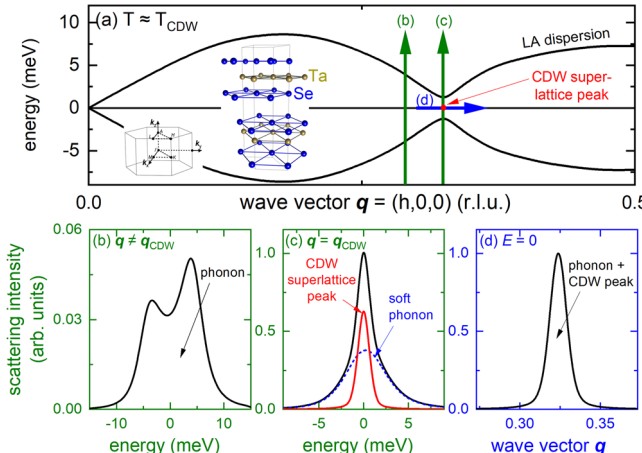

**Fig. 1 | Probing of charge-density-wave order in 2$H$-TaSe$_2$. a** Schematic dispersion (black solid lines) of an acoustic phonon with a soft mode at $\mathbf{q}_{CDW} \approx (1/3,0,0)$ for energy-loss (positive energies) and energy-gain (negative energies). The red dot indicates the position of the corresponding superlattice peak in the ordered phase. Thick vertical (green) and horizontal (blue) arrows illustrate the scans done on the HERIX spectrometer to investigate the phonon softening and superlattice peak formation in 2$H$-TaSe$_2$. Labels **b**–**d** refer to the following panels showing typical results for the corresponding scans in more detail. Insets in (**a**) show the crystal structure of 2$H$-TaSe$_2$ ($P6_3/mmc$, $a = b = 3.44$ Å, $c = 12.7$ Å, #194) and the layout of the Brillouin zone with high symmetry points labeled. **b** Energy scan at $\mathbf{q} \neq \mathbf{q}_{CDW}$. For a realistic picture we convoluted a damped harmonic oscillator function with the pseudo-voigt-like experimental resolution ($\Delta E_{FWHM} = 1.5$ meV). **c** Energy scan at $\mathbf{q} = \mathbf{q}_{CDW}$. The signal from the superlattice peak is approximated by the resolution function whereas the damped phonon is represented by a damped harmonic oscillator function convoluted with the resolution function. The scattering contributions from the superlattice peak [thick (red) solid line] are easily distinguished from the phonon contribution [thick (blue) dashed line]. **d** Momentum scan at zero energy transfer, $E = 0$, across the CDW superlattice peak (red dot in (**a**)). Because of the finite energy resolution and the broad phonon lineshapes, it is not clear how strongly soft phonon mode and superlattice peak contribute to the scattering at zero energy transfer.

softening was not complete, i.e., the energy of the LA mode at $\mathbf{q}_{CDW}$ softened from above 7 meV at room temperature only to 4.5 meV at $T_{CDW}$. However, a resolution-limited, static central peak at zero energy transfer developed already below 150 K, well above $T_{CDW}$ where phonon softening was reported to be strongest. Similar observations, i.e., the rise of a central elastic peak before the corresponding phonon fully softens, had been made earlier in ferroelectric SrTiO$_3$[43–45] and more recently in quasi-1D CDW compounds[22] as well as in cuprates[2]. The origin of the central peak is not fully understood[45] though it was ascribed to ordered fluctuations in some CDW-related materials[2,22]. The report of such an elastic peak rising above the phase transition temperature in 2$H$-TaSe$_2$ is one of the earliest documented, although the authors of the original INS study[38] pointed out that the coarse momentum resolution in their experiments may have obscured a better view of the lattice dynamics. Hence, the lattice dynamical behavior in 2$H$-TaSe$_2$ remains unclear and is highly relevant not only for its intertwined superconducting and charge-ordered phases but also for our understanding of other materials featuring CDW order such as quasi-1D (TaSe$_4$)$_2$I[36], NbSe$_3$[35], ZrTe$_3$[22], and underdoped cuprates[2,3].

Here, we employ inelastic x-ray scattering (IXS) with meV resolution to study the CDW soft phonon mode in 2$H$-TaSe$_2$. The much better momentum resolution of IXS compared to INS enables the detailed study of the soft-mode properties as a function of wave vectors as well as temperature. Combined with the energy resolution of 1.5 meV, IXS enables an unambiguous separation between the static superlattice peak (resolution limited in energy) and in- or quasi-elastic (not resolution limited) scattering from the soft phonon mode. We observe a full phonon softening of the LA phonon mode at $\mathbf{q}_{CDW} = (0.323,0,0)$ and that the static CDW superlattice peak intensity rises strongly only on cooling to temperatures below that of the full phonon softening. This is in contrast to previous results[38] and rules out the central-peak scenario for 2$H$-TaSe$_2$. Yet, the most surprising finding is that the phonon softening occurs at a temperature $T^* = 128.7$ K, i.e., well above $T_{CDW} = 121.3$ K. Analysis of high momentum resolution scans at $E = 0$ reveals that the phase at $T_{CDW} \leq T \leq T^*$ is characterized by lattice fluctuations, observed as the overdamped soft phonon mode, coexisting with static but only medium-range CDW order. Furthermore, we performed ab initio lattice dynamical calculations to investigate the origin of CDW order as well as the role of the CDW soft phonon mode with regard to the increasing superconducting temperatures at high pressure. The results show that both electron–phonon coupling, as well as the Fermi surface topology, have to be taken into account to understand the origin of CDW order in 2$H$-TaSe$_2$. It also seems to be an ideal candidate to investigate the interplay of a CDW quantum critical point and emergent superconductivity relevant to a number of other systems of interest, including cuprates and 122 Ni-based superconductors.

## Results

The strategy of our IXS experiment is outlined in Fig. 1: The LA mode at $\mathbf{q} = (h,0,0)$ shows a pronounced softening close to the temperature of the phase transition at $\mathbf{q}_{CDW}$ (black lines in Fig. 1a) where eventually the CDW superlattice peak (red dot at zero energy transfer) emerges. We performed energy scans at constant momentum transfer [vertical (green) arrows] and momentum scans at zero energy transfer [horizontal (blue) arrows]. Typical results for three different scenarios are sketched in Fig. 1b–d. At $\mathbf{q} \neq \mathbf{q}_{CDW}$, the LA mode can already have a low energy, e.g., $E_{phon} \approx 4$ meV. Taking into account the energy resolution and damping due to electron–phonon coupling, the resulting scan shows broad phonons peaked at $\pm E_{phon}$ and with finite scattering intensity even at zero energy transfer (Fig. 1b). Constant momentum scans at $\mathbf{q} = \mathbf{q}_{CDW}$ and $T \approx T_{CDW}$ typically show a resolution-limited signal corresponding to the rising static CDW superlattice peak (solid (red) line in Fig. 1c) easily distinguishable from the broad scattering from the damped soft phonon mode [dashed (blue) line]. Due to the

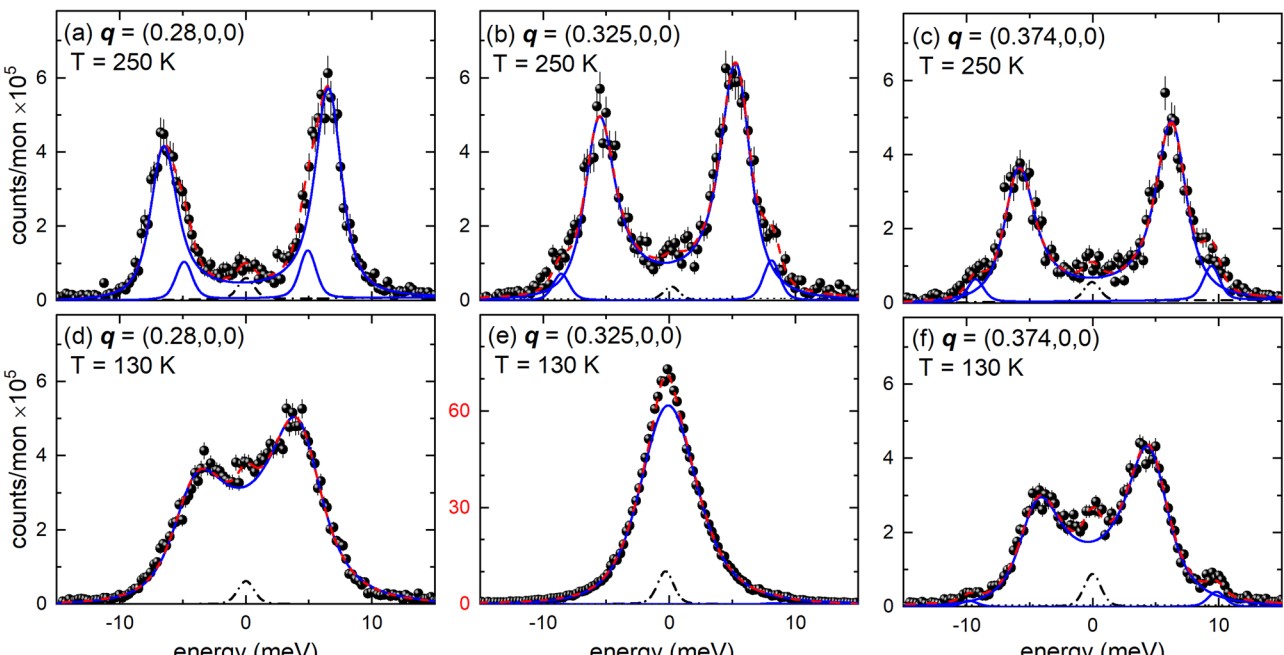

**Fig. 2 | Energy scans of the soft phonon mode in 2H-TaSe₂.** Energy scans taken at **q** ∥ [100] at **a**–**c** T = 250 K and **d**–**f** 130 K. Dashed (red) lines are fits consisting of DHO functions convoluted with the experimental resolution (blue solid lines),

estimated background (straight dotted line) and a resolution limited pseudo-Voigt function for the elastic line (black dash-dotted line). Error bars represent standard deviations. Error bars represent s.d.

small phonon scattering intensities, constant momentum scans employ a relatively broad momentum resolution, $\Delta q = 0.09\,\text{Å}^{-1}$, not suited to investigate the correlation length of the rising CDW super-lattice peak. Therefore, we performed additional momentum scans at zero energy transfer across $\mathbf{q} = \mathbf{q}_{\text{CDW}}$ with high resolution, $\Delta q = 0.018\,\text{Å}^{-1}$. Generally, it is not clear which part of the scattering intensity at zero energy transfer originates from the static elastic peak and which part is due to the (nearly) completely damped LA phonon (Fig. 1d). Yet, the combined analysis of both types of scans allows the unambiguous determination of the temperature dependence of the elastic CDW superlattice peak concomitant with that of the soft phonon mode.

Figure 2 illustrates raw data along the dispersion of the LA mode (Fig. 2a–f) and highlights its strong temperature dependence at $\mathbf{q} = (0.325,0,0)$ [$\approx \mathbf{q}_{\text{CDW}} = (0.323,0,0)$] (Fig. 2b and e). The temperature-dependent energy of the LA mode at $\mathbf{q}_{\text{CDW}}$ (Fig. 3a) shows a clear soft-mode behavior. Surprisingly, the softening is, within our experimental resolution, complete already just below 130 K, well above the reported CDW transition temperature[38] of about 122 K (vertical blue dashed line in Fig. 3a). Fitting a power law of the form $|T - T^*|^\delta$ (red solid line in Fig. 3a) to the temperature dependence of the soft mode for $T \geq 130$ K yields $\delta = 0.32(2)$ and $T^* = 128.7(3)$ K (vertical red dashed line in Fig. 3a). Below $T^*$, the energy of the soft mode stays close to zero. The observed value of $\delta$ is lower than the mean-field one, $\delta_{\text{MF}} = 0.5$, observed e.g. for the CDW transition in the iso-structural 2H-NbSe₂[25]. Since mean-field theory neglects fluctuations of the order parameter, $\delta = 0.32(2)$ indicates the presence of strong order parameter fluctuations in 2H-TaSe₂. We expect reduced fluctuations far away from the transition and, indeed, the soft-mode energies for $T \geq 175$ K follow mean-field behavior, which means that the square of the phonon energies depends linearly on temperature (see Fig. 3b). A linear fit for $T \geq 175$ K extrapolates zero energy for the soft mode at $T \approx 85$ K, which is close to $T_{\text{C-IC}} \approx 88$ K[41,42], the temperature at which 2H-TaSe₂ acquires a commensurate CDW.

Before we discuss the temperature range $T \approx T^*$ in more detail, we present the dispersion and the damping of the LA soft mode in Fig. 3c, d, respectively. We find, within our experimental resolution, a V-like

dispersion and a sharply peaked damping centered at $\mathbf{q}_{\text{CDW}}$ at $T = 130$ K (black spheres in Fig. 3c, d) again in contrast to 2H-NbSe₂, where the LA soft mode displays a much broader, U-shaped dispersion at the respective $T_{\text{CDW}}$[25,46]. Phonon energies of the LA mode rise much faster going away from $q_{\text{CDW}}$ in 2H-TaSe₂ than in 2H-NbSe₂. A V-like dispersion (Fig. 3c) is expected when the EPC of the soft mode is dominated by a Fermi surface (FS) nesting. In that case, the strong increase of electronic decay channels at the nesting wave vector leads to sharply localized phonon anomalies. The dispersion of 2H-TaSe₂ is sharper than in 2H-NbSe₂, in which FS nesting was shown to be completely absent[47], but not as sharp as in some quasi-1D compounds, such as KCP[21] and ZrTe₃[22]. This is in line with results from angle-resolved photoemission spectroscopy (ARPES) revealing a moderately strong FS nesting in 2H-TaSe₂[48]. The range of wave vectors over which the LA mode shows significant renormalization on cooling to $T^*$ is 0.032(10) r.l.u. along the [100] direction (half-width of phonon anomaly, Fig. 3c) indicating a short coherence length of the CDW order parameter of $\xi_0 = 17 \pm 4$ Å, in reasonable agreement with estimates from specific heat $\xi_0 \approx 14$ Å[49].

Figure 4 shows energy scans at $\mathbf{q}_{\text{CDW}}$ taken for temperatures $T_{\text{CDW}} \leq T \leq T^*$ along with an analysis of the temperature dependence of the observed static CDW superlattice peak. The results show that the CDW superlattice-peak intensity (dash-dotted red lines in Fig. 4a–c) increases in this temperature range by a factor of 10. Yet, the spectrum at $T = 122.5$ K still shows similar spectral weights of the CDW super-lattice peak (dash-dotted red line) and the soft phonon mode (solid blue line in Fig. 4c). Cooling further by only $\Delta T = 1$ K, the scattering at zero energy jumps by a factor of 20 (inset of Fig. 4c) and we will see further below that this jump marks the onset of long-range CDW order. However, the strong increase of the superlattice peak prohibits further analysis of in- or quasi-elastic scattering at lower temperatures with the given energy resolution, because the much weaker quasi-elastic scattering is hidden under the tail of the elastic peak. The temperature dependences of the soft mode's energy, its intensity, and that of the CDW superlattice peak are summarized in Fig. 4d, e, respectively. The intensity of the soft phonon mode dominates the spectra except for close to and, of course, below $T_{\text{CDW}}$. Nevertheless, our results clearly

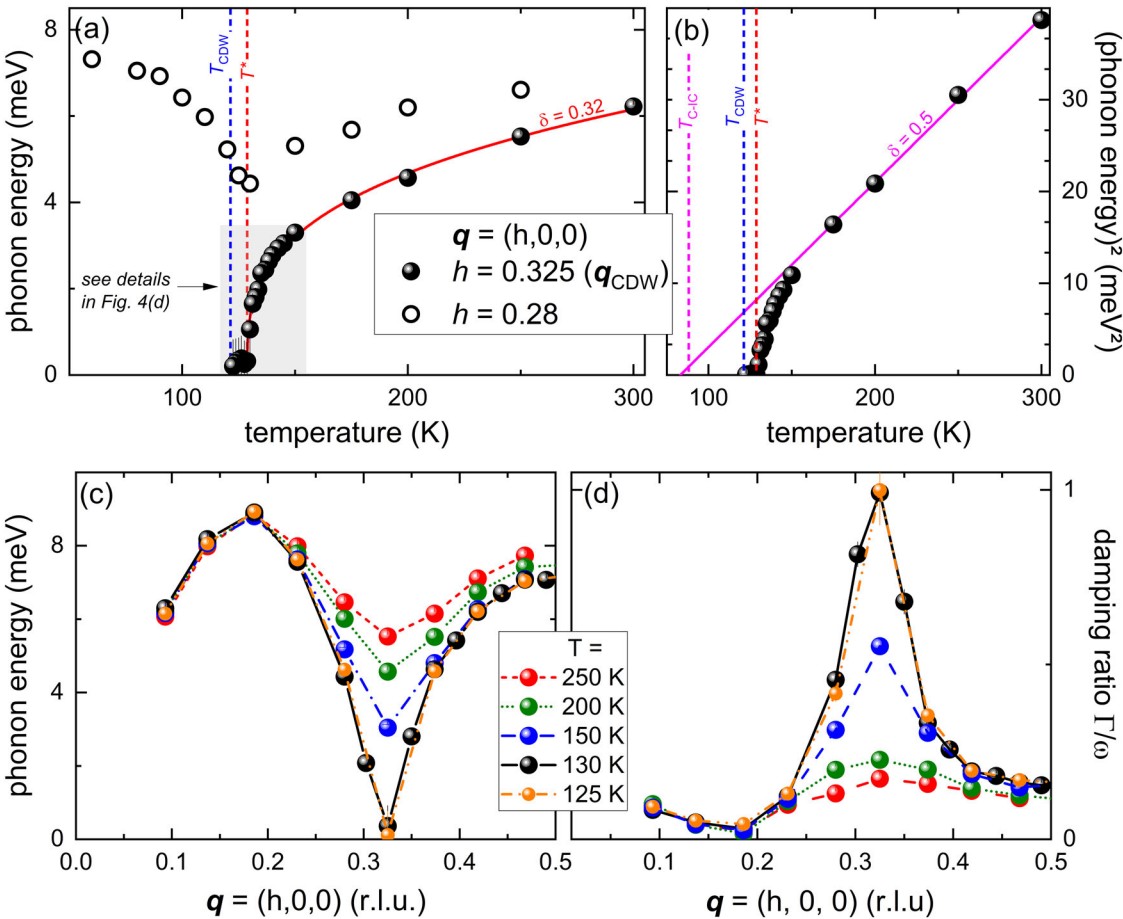

**Fig. 3 | Temperature and momentum dependence of the soft phonon mode in 2H-TaSe₂. a** Temperature dependence of the energies of the LA phonon modes at $\mathbf{q} = (0.325,0,0)$ (=$\mathbf{q}_{CDW}$, spheres) and $\mathbf{q} = (0.28,0,0)$ (open circles). The line is a power law fit to the data for $\mathbf{q} = \mathbf{q}_{CDW}$ and $T \geq 130$ K of the form $|T-T^*|^{\delta}$ yielding $T^* = 128.7$ K (vertical red dashed line) and $\delta = 0.32 \pm 0.02$ (data in gray-shaded box are shown in more detail in Fig. 4d). The vertical blue dashed line indicates $T_{CDW} = 121.3$ K. **b** Square of the phonon energy at $\mathbf{q} = \mathbf{q}_{CDW}$ as function of temperature. The solid line is a linear fit corresponding to $\delta = 0.5$ in (**a**) for $T \geq 175$ K. $T_{IC-C} \approx 88$ K indicates the transition temperature into the commensurate CDW phase (see inset in Fig. 5c). **c** Dispersion and **d** damping ratio $\Gamma/\omega$ of the LA mode along $\mathbf{q} = (h,0,0)$ for $250$ K $\geq T \geq 125$ K. Lines are guides to the eye. Error bars represent s.d.

show that a small static CDW superlattice peak is already present at $T \approx 130$ K. The increase of the CDW superlattice peak intensity accelerates sharply below $T^*$ (vertical red dashed line in Fig. 4e) before it jumps by a factor of 20 crossing $T_{CDW}$ (see also inset in Fig. 4c).

Our results identify a precursor region at $T_{CDW} \leq T \leq T^*$ characterized by the simultaneous presence of soft-mode lattice fluctuations and static CDW order. A critically damped phonon indicates that there is quasi-zero energy needed for small lattice distortions that are symmetry-related to the phonon oscillation pattern. On the other hand, the intensity of a superlattice peak is proportional to the square of the order parameter, i.e., the atomic displacement from the high-temperature equilibrium position. Thus, the presence of a weak CDW superlattice peak indicates small but finite displacements. A possible interpretation is that there exist small domains of static CDW order with a finite correlation length $\xi_{corr}$, in a matrix of material with critical lattice fluctuations. On cooling, one would expect that $\xi_{corr}$ increases and finally diverges at the temperature at which the sample reaches long-range CDW order, i.e., at $T_{CDW}$.

However, the momentum resolution employed for energy scans ($\Delta q = 0.09$ Å⁻¹) is too coarse for a detailed analysis of the CDW correlation length $\xi_{corr}$. Therefore, we performed momentum scans at constant energy transfer of $E = 0$ (indicated by the horizontal blue arrow in Fig. 1a and illustrated in Fig. 1d) with an improved momentum resolution of $\Delta q = 0.018$ Å⁻¹ (see the "Methods" section and Supplemental Note 2). Raw data at selected temperatures are shown in Fig. 5a.

Data at $T = 122$ K (red spheres in Fig. 5a) show the rise of scattering intensity for a broad momentum range $\mathbf{q} = (0.3–0.35,0,0)$ on cooling. Just 1 K below, at $T = 121$ K, we find a more than twenty-fold increase of the scattering intensity centered at $\mathbf{q}_{CDW} = (0.323,0,0)$. On further cooling the intensity increases further and the peak position moves towards the commensurate wave vector $\mathbf{q}_{CDW,C} = (1/3,0,0)$ illustrated by the data for $T = 80$ K (orange spheres in Fig. 5a). From a peak fit we obtain the detailed temperature dependences of the integrated intensity (Fig. 5b), the peak position $\mathbf{q}_{CDW}$ (Fig. 5c) and its line width $\Gamma_{exp,FWHM}$ (Fig. 5d). The integrated peak intensity follows a power law of the form $|T-T_{CDW}|^{2\beta}$ for $T \leq 121$ K with $T_{CDW} = 121.3(2)$ K and $\beta = 0.21(1)$ (red line in the inset of Fig. 5b). On further cooling, the intensity continues to increase and $\mathbf{q}_{CDW}$ shifts to the commensurate value $\mathbf{q}_{CDW,C} = (1/3,0,0)$ at temperatures close to $T_{C-IC} \approx 88$ K[42] (inset in Fig. 5c). $\Gamma_{exp,FWHM}(T)$ shows that peaks are resolution limited for data taken at $T \leq 121$ K. A power law fit of the form $(T-T_{CDW})^{\delta}$ to $\Gamma_{exp,FWHM}(T)$ for $T \geq 122$ K corroborates the transition temperature to long-range CDW order $T_{CDW} = 121.3(2)$ K in good agreement with the literature[38,41,42,49]. A more detailed analysis which takes into account the experimental resolution and estimates the phononic background (see Fig. S2 and Supplemental Note 2) shows that the correlation length $\xi_{corr}$ of the static CDW domains in the precursor region increases along [100] on cooling from 100 to 200 Å just above $T_{CDW}$ (black spheres in Fig. 5e). Values along [010] are essentially the same while the correlation length along [001] is 4–5 times smaller. Thus, the precursor region

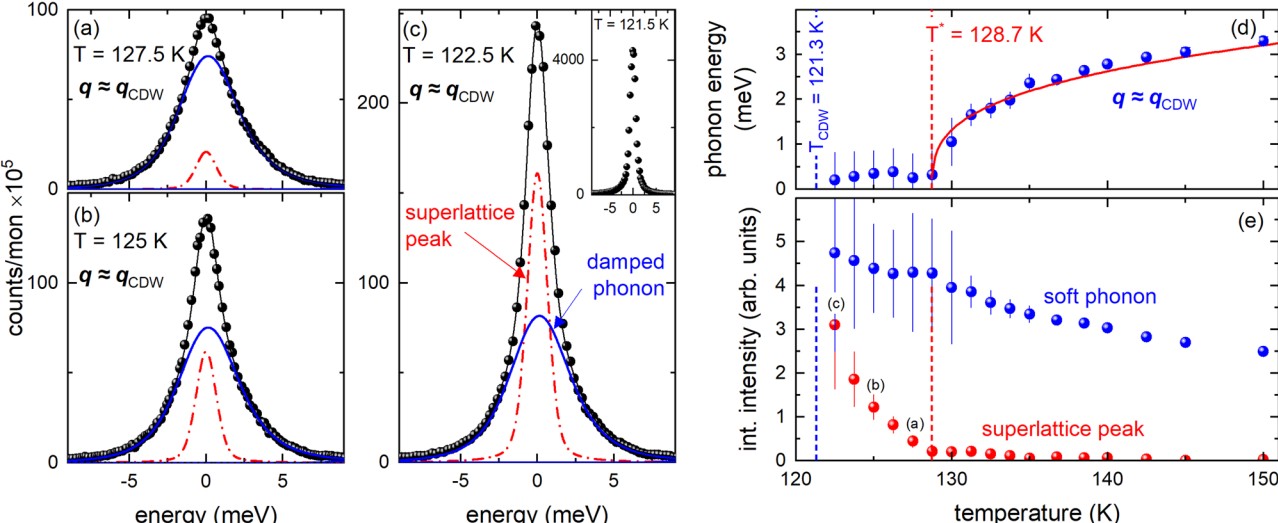

**Fig. 4 | Evolution of elastic and inelastic scattering at $T_{CDW} \leq T \leq 150\,K$. a–c** Energy scans taken at $\mathbf{Q} = \boldsymbol{\tau} + \mathbf{q}$ with $\boldsymbol{\tau} = (3,0,1)$ and $\mathbf{q} = \mathbf{q}_{CDW}$ [=(0.325,0,0)] at various temperatures $122.5\,K \leq T \leq 127.5\,K$, i.e., $T_{CDW} < T < T^*$. Thin solid (black) lines are fits consisting of DHO functions convoluted with the experimental resolution (blue solid lines), constant background, and a resolution-limited pseudo-Voigt function for the rising CDW superlattice peak (red dash-dotted line). The inset in **c** shows data taken at $T = 121.5\,K$ ($\approx T_{CDW}$). Here, no analysis of inelastic scattering is possible because of the increase of the CDW superlattice peak by more than a factor of 20.

**d** Temperature-dependent energy of the CDW soft phonon mode (blue spheres). The red solid line represents the same power-law fit as shown in Fig. 3a. **e** Temperature-dependent integrated intensities of the CDW superlattice peak (red spheres, see also red dash-dotted lines in (**a–c**)) and the soft phonon mode (blue spheres, see also blue solid lines in (**a–c**)). Small letters, **a–c**, reference panels showing the corresponding IXS raw data. Vertical blue and red dashed lines (in **d** and **e**) denote $T_{CDW} = 121.3\,K$ and $T^* = 128.7\,K$, respectively. Error bars represent s.d.

in 2*H*-TaSe₂ is characterized by medium-range-sized CDW domains, which only form a long-range ordered state at $T \leq T_{CDW}$.

The precursor region has not been observed previously by other techniques. INS experiments[38,41] did not have sufficient momentum resolution while XRD[42] was energy integrated and so could not distinguish between static CDW and soft-mode intensity contributions. We performed synchrotron-based ARPES to cross-check results on our own samples with previous reports[40]. In particular, we looked for quantities previously used as a measure of the onset of CDW order such as a pseudo-gap[40] and the evolution across $T^*$. Measurements were done with an incident photon energy of 80 eV (resolution <10 meV) over a temperature range $132\,K \geq T \geq 113\,K$, i.e., cooling across $T^*$ and $T_{CDW}$. Technical details and detailed results of our analysis are given in Supplemental Note 3. Here, we summarize, that—in agreement with previous reports—the electronic band structure exhibits a pseudo-gap (as defined in ref. 40, see Supplemental Note 3 and Fig. S3 for more details), which increases slowly on cooling in the upper part of the temperature range (open squares in Fig. 5e). The increase of the pseudo-gap on cooling accelerates by a factor 4 at lower temperatures and linear fits within the two temperature regions (dashed lines in Fig. 5e) intersect very close to $T_{CDW} = 121.3\,K$. However, no particular response is detectable on crossing $T^*$. The presence of the pseudo-gap in 2*H*-TaSe₂ has been reported up to nearly room temperature[48] as well as in the high-temperature phase of 2*H*-NbSe₂[50]. The latter report found that the pseudo-gap phase in 2*H*-NbSe₂ at $T > T_{CDW}$ is related to incoherent CDW fluctuations. Long-range CDW order only sets in once phase coherence is established at $T < T_{CDW}$. On the other hand, electrons adjust quasi-instantaneously to lattice motions. It is argued in the case of 2*H*-NbSe₂[50] that fluctuations always yield a pseudo-gap irrespective of the time-scale of the fluctuations. Therefore, we argue that electronic degrees of freedom do not distinguish between lattice fluctuations with a finite lifetime, i.e., the overdamped soft mode, and static correlations but follow the evolution of the energy-integrated signal as probed by standard x-ray diffraction[42]. Consequently, the pseudo-gap is rather insensitive to $T^*$ but responds to the large increase of scattering intensity on cooling below $T_{CDW}$.

We performed calculations based on density-functional perturbation theory (DFPT, for details see the "Methods" section and Supplemental Note 4) to investigate the microscopic origin of CDW order and the impact of the CDW soft mode regarding the strongly increasing superconducting transition temperature at high pressure[31]. Results based on ambient-pressure lattice constants[38] correctly assess the CDW structural instability as an anomaly of the longitudinal acoustic phonon propagating along the Γ–M direction (thick solid line in Fig. 6a). The soft mode features imaginary phonon frequencies where the leading instability around $\mathbf{q} \approx (0.35,0,0)$ is close to the experimentally observed $\mathbf{q}_{CDW}$. We further calculated the electronic contribution to the phonon linewidth

$$\gamma_{EPC,DFT}^{\mathbf{q}} = \pi \omega_{\mathbf{q}\lambda} \sum_{\mathbf{k}} \left| g_{\mathbf{k+q,k}}^{\mathbf{q}\lambda} \right|^2 \delta(\epsilon_{\mathbf{k}} - \epsilon_F)\delta(\epsilon_{\mathbf{k+q}} - \epsilon_F) \quad (1)$$

where $g_{\mathbf{k+q,k}}^{\mathbf{q}\lambda}$ is the EPC matrix element for a phonon mode $\lambda$ at a wave vector $\mathbf{q}$, $\omega_{\mathbf{q}\lambda}$ is the mode's energy and $\sum_{\mathbf{k}} \delta(\epsilon_{\mathbf{k}} - \epsilon_F)\delta(\epsilon_{\mathbf{k+q}} - \epsilon_F)$ gives the number of electronic states at the Fermi level connected by the phonon wave vector $\mathbf{q}$, i.e., the electronic joint density-of-states (eJDOS).

The momentum dependence of $\gamma_{EPC,DFT}^{\mathbf{q}}$ of the CDW soft phonon mode is in good qualitative agreement with our experimental observations (solid line and symbols in Fig. 6b). The fact that the experimental values are significantly larger than the calculated ones has been already observed in several other IXS measurements in CDW systems[25,51,52]. While anharmonic effects close to the structural phase transition which are not included in our quasi-harmonic calculations might play a role, the finite momentum resolution in concert with steeply dispersing phonon modes will also lead to increased linewidths near the phase transition temperature (see Supplemental Note 2 for more details). For 2*H*-TaSe₂, the eJDOS for electronic band transitions mediating strong coupling to the CDW soft phonon mode shows a clear maximum near $\mathbf{q}_{CDW}$ (solid line in Fig. 6c, for more details see Figs. S4–S6 and Supplemental Note 4). The peak in the eJDOS is not very sharp in agreement with the reported moderately strong FS nesting in 2*H*-TaSe₂[48]. A more detailed analysis (see Figs. S4–S6 and

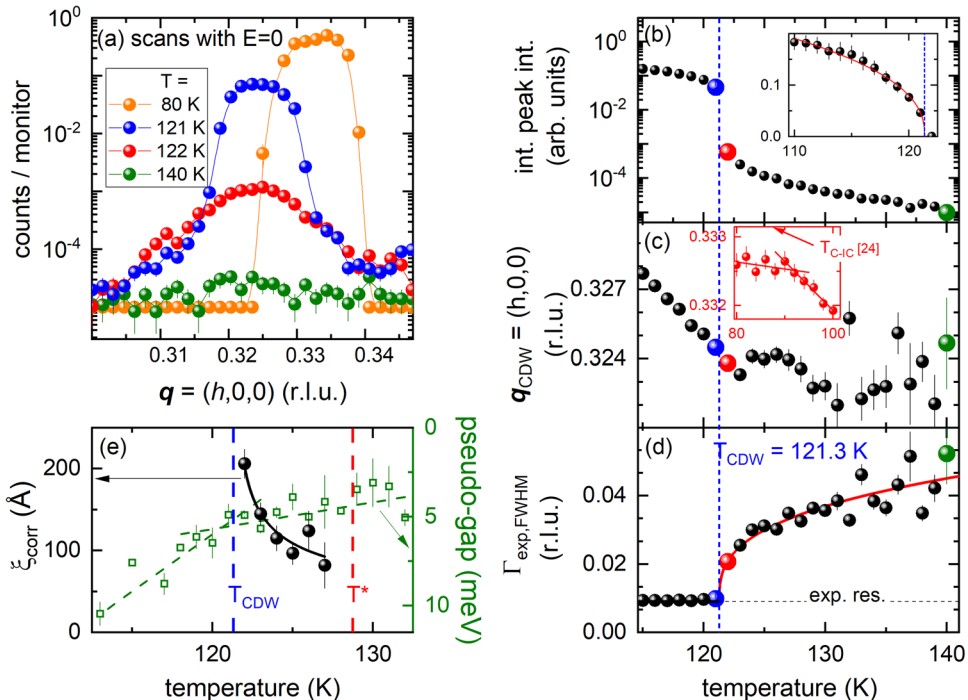

**Fig. 5 | High-resolution momentum scans at $E = 0$ and temperatures $T \leq 140$ K.**
**a** Momentum scans at zero energy transfer along the [100] direction, i.e., across
$q_{CDW}$ for temperatures above and below the CDW transition. Data were taken in a
high-momentum resolution setup (see Supplemental Note 2) to resolve the evo-
lution of $q_{CDW}$ on cooling below $T_{CDW}$. **b–d** Temperature-dependent **b** integrated
intensity, **c** position $q_{CDW}$ and **d** line width $\Gamma_{exp,FWHM}$ of peak fits to the momentum
scans at zero energy transfer. Color-coded dots denote temperatures with data
shown in (**a**). The inset in **b** shows the integrated intensity on a linear scale along
with a power law fit (red line) of the form $|T - T_{CDW}|^{\delta}$ yielding $T_{CDW} = 121.3(2)$ K. The
inset in **c** focuses on temperatures around $T_{C\text{-}IC} \approx 88$ K[42]. Lines are linear fits to the
range below and above $T_{C\text{-}IC}$. The solid red line in **d** is another power law fit to the

corresponding data for $T \geq 122$ K and yields $T_{CDW} = 121.3(2)$ K, indicated by the
vertical blue dashed line. The dashed horizontal line in **d** denotes the experimental
momentum resolution. **e** The temperature-dependent correlation length $\xi_{corr}$
(spheres) of the static scattering was obtained by subtracting the data at $T = 129$ K
($\approx T^*$) as background and analyzing the linewidth of the static scattering rising below
$T^*$ (see text, Supplemental Note 2 and Fig. S2). The solid line is a guide to the eye.
The vertical blue and red dashed lines denote $T_{CDW} = 121.3$ K and $T^* = 128.7$ K,
respectively. Green squares denote the temperature-dependent pseudo-gap
deduced from ARPES measurements (see text, Supplemental Note 3 and Fig. S3).
Dashed lines are linear fits to the data for $T < T_{CDW}$ and $T > T_{CDW}$. Error bars
represent s.d.

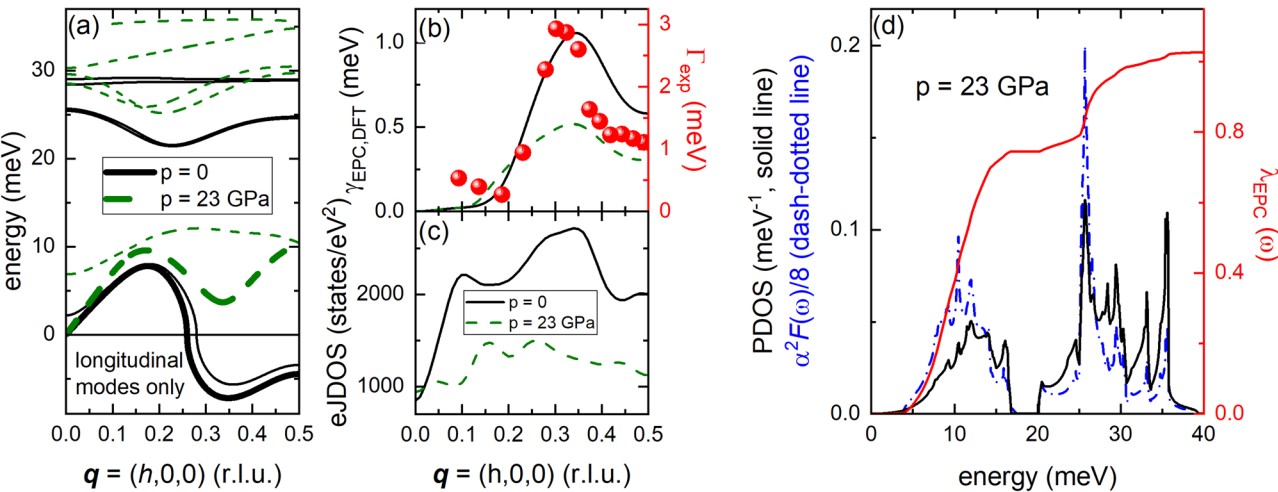

**Fig. 6 | Ab-initio lattice dynamical calculations. a** Calculated dispersion for
longitudinal phonons along the $\Gamma$-M direction in $2H$-TaSe$_2$—performed in the hex-
agonal unit cell, $P6_3/mmc$. Imaginary phonon energies denote an unstable struc-
ture and are shown as negative values. **b** Calculated electronic contribution to the
phonon linewidth $\gamma_{EPC,DFT}$ (FWHM) of the LA soft phonon mode (thick solid and
dashed lines in (**a**)). Symbols represent the linewidth (FWHM) of the DHO function
approximated to the IXS data (right-hand scale, error bars represent s.d.).
**c** Calculated electronic joint density-of-states (eJDOS) for the band transition

mediating EPC of the soft mode (see Figs. S4–S6 and Supplemental Note 4). Solid
lines denote calculations for ambient-pressure lattice constants ($a = b = 3.436$ Å,
$c = 12.7$ Å). Dashed lines represent calculations based on the high-pressure lattice
constants ($a = b = 3.242$ Å, $c = 11.09$ Å, see text). **d** Calculations for the high-pressure
lattice constants showing the phonon density-of-states (PDOS), Eliashberg function
$\alpha^2 F(\omega)$, and electron–phonon coupling constant $\lambda_{EPC}$ as function of phonon
energy.

Supplemental Note 4) explains that a previously reported[19], and by us reproduced, nesting feature at $\mathbf{q}_{nest} = (1/3, 1/3, 0)$ is not related to any significant EPC and, thus, irrelevant for the LA soft phonon mode. Thus, $2H$-TaSe$_2$ is an excellent example where the CDW properties are linked to an interplay of momentum-selective EPC matrix elements and FS topology. Regarding the superconducting properties at ambient pressure, it is expected that the EPC of the low-energy phonons is mostly gapped out by the opening of the CDW energy gap in the electronic band structure[39], and, thus, cannot mediate superconductivity at $T \ll T_{CDW}$. On the other hand, the coupling of the optic branches with energies above 20 meV is relatively small. Estimates for a corresponding superconducting transition temperature are zero or very small (see Supplemental Note 4) consistent with the experimentally observed $T_{sc} \approx 0.1$ K at ambient pressure[31].

Interestingly, calculations based on lattice constants obtained for $p = 23$ GPa[53] [$T_{c,max}$ ($p = 23$ GPa) $= 8.2$ K][31] predict a stable phonon spectrum for the hexagonal structure of $2H$-TaSe$_2$ although the anomaly around $\mathbf{q}_{CDW}$ is still clearly visible (thick dashed line in Fig. 6a). The stabilization of the LA mode at high pressure is accompanied by a strong reduction of its linewidth $\gamma_{EPC,DFT}$ (dashed line in Fig. 6b) and eJDOS (dashed line in Fig. 6c). In particular the peak in the eJDOS near $\mathbf{q}_{CDW}$ is flattened suggesting that the pressure-induced changes of the electronic band structure play an important role in the suppression of charge order at high pressure in $2H$-TaSe$_2$. Thus, we can analyze the EPC of $2H$-TaSe$_2$ when the CDW ground state is suppressed at high pressure (Fig. 6d). We find that more than 70% of the overall EPC constant $\lambda_{EPC} = 1.03$ originates from phonons below 15 meV (red solid line in Fig. 6d). The estimated value of $T_{c,DFT} = 10.6$ K is in reasonable agreement with the experimentally observed $T_{c,exp} = 8.2$ K at $p \approx 23$ GPa[31]. We conclude that the EPC of the low-energy phonons in $2H$-TaSe$_2$ including the CDW soft phonon mode can drive the large increase of the superconducting transition temperature at high pressure.

## Discussion

The presented IXS investigation puts the CDW transition in $2H$-TaSe$_2$ in a new light in that there is a full phonon softening. Our analysis shows that the tail of the scattering above T$_{CDW}$ (Fig. 5b), observed in synchrotron XRD[42] and previous neutron scattering work[38,41], is dominated by scattering from the soft LA mode (see Fig. 4e). Thus, we can rule out the central-peak scenario for $2H$-TaSe$_2$ as it was observed, e.g., in SrTiO$_3$[43–45], and other CDW compounds[2,3,22,35,36].

A new striking feature of the CDW transition in $2H$-TaSe$_2$ is the precursor region for $T_{CDW} = 121.3$ K $\leq T \leq T^* = 128.7$ K defined by the presence of an overdamped phonon mode and the slow increase of the static elastic line. Both features appear at $\mathbf{q}_{CDW}$. Thus, the characteristic features of the precursor region are not competitive but rather precursor-like with regard to the CDW phase at $T \leq T^*$. The presence of an overdamped phonon over a finite temperature range bears similarities to a liquid-like scenario for the soft phonon mode[54,55]. For instance, Ag-based argyrodites, which are promising candidates for thermoelectric applications, feature a structural phase transition temperature $T_s$ at which a phonon softens on heating towards $T_s$. This soft optical phonon mode becomes and stays over-damped at temperatures $T \geq T_s$ and is linked to the diffusion of Ag ions in these materials[54,56]. Normally, the transition from a full soft phonon mode to a long-range CDW order occurs at the same temperature, e.g., in $2H$-NbSe$_2$[25], $1T$-VSe$_2$[28] or the rare-earth tritellurides[51,57]. A possible reason for the anomalously wide temperature range, $\Delta T \approx 7.4$ K, in which this evolution happens might be strong fluctuations that prohibit a direct transition to a CDW long-range ordered phase. This transition only occurs at the lower end of $\Delta T$, i.e., $T_{CDW}$ accompanied by a large increase of the intensity of the, then resolution limited, superlattice peak. It is worth noting that the 20-fold increase of the elastic scattering within $\Delta T = 1$ K from above to below $T_{CDW}$ (see Fig. 4c and inset) is much more abrupt than in similar measurements in $2H$-NbSe$_2$. In the latter compound, the rise is rather gradual and the intensity is not even doubled on cooling $\Delta T = 1$ K at any temperature close to or below $T_{CDW,NbSe2}$[25]. Similarly, specific heat data for $2H$-TaSe$_2$ shows a sharp peak at $T_{CDW}$ with a width of $<1$ K[49], whereas equivalent data for $2H$-NbSe$_2$ feature a width of more than 5 K[58]. It requires additional theoretical modeling which is able to include the effects of fluctuations to understand the details of the CDW transition in $2H$-TaSe$_2$ beyond the scope of the work reported here.

Our results regarding the elastic superlattice peak compared to the weak pseudo-gap in the electronic band structure demonstrate that structural order precedes full electronic order in $2H$-TaSe$_2$. Early specific heat measurements in $2H$-TaSe$_2$[49] as well as theoretical considerations[59] already suggested a dominant lattice, i.e., phonon contribution to the phase transition entropy on entering the incommensurate CDW phase at $T_{CDW}$. More recently, CDW order in quasi-1D ZrTe$_3$ has been shown to feature a structural transition at the typically reported $T_{CDW} = 63$ K while charge modulations only become long-range ordered below $T_{LO} = 56$ K[60]. The authors argued that Friedel oscillations seeded by impurities produce the short-range charge modulations at $T_{LO} \leq T \leq T_{CDW}$, whereas only the long-range ordered modulations below $T_{LO}$ bear the true CDW nature. A similar scenario in $2H$-TaSe$_2$ might explain the weak electronic response to the transition at $T_{CDW}$ (see Fig. 5e, 8 and Supplemental Note 3 and refs. 39,40) and a strong one at the onset of commensurate order at $T_{C-IC}$[40,48]. Yet, lattice dynamics in ZrTe$_3$[22] are qualitatively different from those in $2H$-TaSe$_2$ and a more detailed comparison is necessary to understand these common characteristics. Overall, $2H$-TaSe$_2$ seems to be between mean-field-like $2H$-NbSe$_2$ and quasi-1D ZrTe$_3$ in terms of the strength of FS nesting and the impact of fluctuations. Thus, the study of equally important electronic and lattice degrees of freedom in $2H$-TaSe$_2$ presents an interesting topic for future research.

Discussions on the interplay of CDW order and superconductivity have been revitalized by the discovery of CDW states in cuprates[2,3,61]. More recently, a sixfold increase of the superconducting transition temperature $T_{sc}$ was reported for Ba$_{1-x}$Sr$_x$Ni$_2$As$_2$, a non-magnetic sister compound of the 122 iron-based superconductors, featuring a CDW state suppressed near the maximum $T_{sc}$ value[62]. Inelastic x-ray scattering measurements of the parent compound, BaNi$_2$As$_2$, found a full phonon softening at the incommensurate CDW ordering wave vector well above $T_{CDW}$[61,63,64] similar to the precursor region observed by us in $2H$-TaSe$_2$. Interestingly, one report interpreted these findings for BaNi$_2$As$_2$ as a gradual slowing down on cooling of the already fully softened phonon mode before they coalesce into the static long-range CDW order[64]. This is not comparable with our observations for $2H$-TaSe$_2$ where we can always clearly distinguish between the resolution-limited elastic line and the soft phonon mode (see Fig. 4a–c). Furthermore, it is argued for BaNi$_2$As$_2$[64] that the intense lattice fluctuations above T$_{CDW}$ can be responsible for the strong increase of $T_{sc}$ when static CDW order is suppressed. However, Ba$_{1-x}$Sr$_x$Ni$_2$As$_2$ features a 1$^{st}$ order triclinic phase transition just below the onset of CDW order[65,66] and both structural transitions seem to be closely related and are suppressed near the same critical doping[62]. Thus, it will be difficult to establish the presence of a quantum critical point, let alone investigate CDW fluctuations and their impact on superconductivity. Here, we argue that $2H$-TaSe$_2$ under pressure is likely a good candidate to study the soft phonon mode near a CDW quantum critical point. No other structural transition is reported[67] and $T_{sc}$ increases by more than a factor of eighty[31]. Furthermore, the here-reported full phonon softening at ambient pressure necessitates a revision of discussions on the nature of the CDW transition at ambient and finite pressure[31]. Combined with our calculations for high-pressure lattice constants (Fig. 6) and estimated $T_{sc}$ of 10.6 K, our observations suggest that the high-pressure CDW phase might remain 2nd order and can feature a CDW quantum critical point. This would be a rare opportunity since the relation in other CDW compounds often turned out to be less clear

(e.g., $2H$-NbSe$_2$[27,68], $1T$-TiSe$_2$[12,29,69,70], ZrTe$_3$[71–73]). Intercalated $2H$-Pd$_x$TaSe$_2$ also seems to be an interesting candidate material[74]. Here, x-ray diffraction measurements are necessary to unambiguously determine the CDW phase diagram because the reports on $1T$-TiSe$_2$ showed that x-ray diffraction is more accurate than resistivity in determining the CDW phase diagram when CDW order is suppressed by pressure[69,75] or intercalation[29,30].

## Methods

*Inelastic x-ray scattering* experiments were carried out at the 30-ID beamline, HERIX spectrometer[76], at the Advanced Photon Source, Argonne National Laboratory. The sample was a single crystal grown at the University of Kiel and resistivity measurements on a sample from the same growth batch indicate a CDW transition of $T_{CDW,resist.} = (121.7 \pm 0.5)$ K (see Fig. S1 and Supplemental Note 1). All x-ray measurements presented in this work (Figs. 2–5, S2) were done in the same experiment on the HERIX spectrometer. Phonon excitations measured in constant-momentum scans were approximated by damped harmonic oscillator (DHO) functions[77] convoluted with a pseudo-Voigt resolution function [full-width at half-maximum (FWHM) = 1.5 meV, Lorentz ratio = 0.4]. The resolution function was further used to approximate resolution-limited elastic scattering at zero energy transfer. Measurements were done at scattering wave vectors $\mathbf{Q} = \boldsymbol{\tau} - \mathbf{q}$, where $\boldsymbol{\tau}$ is a reciprocal lattice vector and $\mathbf{q}$ is the reduced wave vector in the first Brillouin zone. Wave vectors are expressed in reciprocal lattice units (r.l.u.) ($2\pi/a, 2\pi/b, 2\pi/c$) with the lattice constants $a = b = 3.44$ Å and $c = 12.7$ Å of the high-temperature hexagonal unit cell (#194). All measurements were done in the Brillouin zone adjacent to $\boldsymbol{\tau} = (3,0,1)$. Throughout the text, results are presented in reduced wave vectors $\mathbf{q} = \boldsymbol{\tau} - \mathbf{Q}$. Measurements were done at constant energy or constant momentum transfer. For the former, we employed the best momentum resolution possible on the HERIX spectrometer ($\Delta q = 0.018$ Å$^{-1}$) by decreasing the effective size of the backscattering analyzers to a circular diameter of 18 mm (FWHM) compared to 95 mm opening in energy scans at constant momentum transfer ($\Delta q = 0.09$ Å$^{-1}$). For more details on the HERIX setup and data analysis see Supplemental Note 2.

*Angle-resolved photoelectron spectroscopy* measurements were performed at the Bloch endstation of the R1 synchrotron at the MAXIV institute in Lund, Sweden. More experimental details and results are given in Supplemental Note 3.

*Ab-initio lattice dynamical calculations* based on density-functional-perturbation-theory (DFPT) were performed in the framework of the mixed basis pseudopotential method[42]. The exchange-correlation functional was treated in the local-density approximation (LDA). Spin−orbit interaction was taken into account consistently. More details are given in Supplemental Note 4.

## Data availability

Data used for figures are available at ref. 78. More detailed information including details of the ab-initio calculations is available from the corresponding author upon request.

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

## Acknowledgements

X.S. was supported by the Helmholtz-OCPC Postdoc Program. S.R. was supported by the Materials Sciences and Engineering Division, Office of Basic Energy Sciences, U.S. Department of Energy. B.S., M.d.R.C., and C.M. acknowledge financial support from the Swiss National Science Foundation (SNSF) Grant No. P00P2 170597. M.d.R.C. was supported by grant #2020/13701-0, São Paulo Research Foundation FAPESP. This research used resources from the Advanced Photon Source, a U.S. Department of Energy (DOE) Office of Science User Facility operated for the DOE Office of Science by Argonne National Laboratory under Contract No. DE-AC02-06CH11357. The authors gratefully acknowledge MAX IV Laboratory for time on Beamline Bloch under Proposal 20200293, as well as the support from C. Polley and G. Carbone.

## Author contributions

Inelastic x-ray scattering: X.S., A.A.H., A.H.S., B.M., S.R., F.W.; photoemission spectroscopy: B.S., M.d.R.C., C.M.; theory: R.H., R.H., F.W.; resistivity: M.F.; data analysis: X.S., B.S., C.M., F.W.; sample growth: K.R.; manuscript: X.S., C.M., S.R., F.W.; project coordination: F.W.

## Funding

## Competing interests

The authors declare no competing interests.
