## [Peer Review File · Nature Communications]

Reviewers' Comments:

Reviewer #1:

Remarks to the Author:

Report on the manuscript 'Precursor phase with full phonon softening above the CDW phase transition in 2H-TaSe2'

The work by X. Shen et al report a comprehensive lattice dynamics study of the 2D layered material TaSe2 by means of inelastic x-ray scattering and ARPES. The authors find, in contrast with the previous neutron data taken in the 70's, a complete phonon softening at a temperature higher than the CDW transition. The softening is characterised by a critically damped phonon with a critical exponent lower than the mean field value, highlighting the role of the high temperature charge fluctuations. A detailed discussion of the results in a broad context of TMDs and CDW systems is also included. Finally, the main conclusion is that the Fermi surface nesting scenario is the main driving force of the CDW, unlike other TMDs like VSe2 and NbSe2.

The paper is well written, the experiments were carefully performed and the data analysed. I believe that the manuscript deserves to be published in Nat. Comm. as it brings new information to understand the origin of TMDs and CDWs in general. However, I think that the papers can be improved:

- The authors say that the FS scenario seems to drive the phase transition, instead of EPI. This seems or does not seem to be the case. The softening of the phonon in momentum space is not so sharp as in ZrTe3, although not so broad as the NbSe2, but it still develops 'some broadening'. I ask the authors to quantify the FS, i.e. real and imaginary parts of the electronic susceptibility and momentum dependence EPI and check whether 'peaks' appear and how broad they are. I understand that these calculations can be rather straightforward (although computationally expensive) given the expertise of the group. These calculations will also give a push to the manuscript in Nat. Comm.

- The phonon softens at $T > T_{CDW}$ (7-9 K above T_{CDW}), then, is the softening a precursor of the CDW phase at a different q ? or, competing? I do not find a proper discussion on the origin of the sharp increase of the elastic line at T_{CDW} . If not phonon driven, is this order-disorder, chemical bond instability type,...

Finally, I would suggest the authors to improve the introduction and make it more Nature style. The intro is too narrow, focusing only in TaSe2 and TMDs. I would suggest to broaden it: cuprates, kagomes,... Note that in the cuprates the stripe order seems to lock the high temperature charge fluctuations to the commensurate value. Also, in the Kagomes, the dynamics of CDW does not show a soft mode, but the elastic line increases sharply defining an order-disorder phase transition...

With addressing these comments, I recommend the paper to be published in Nat Comm.

Reviewer #2:

Remarks to the Author:

In the present paper, X. Shen et al. employ inelastic x-ray scattering to solve a decades-old puzzle about the soft phonon mode at the CDW ordering vector in the layered transition-metal chalcogenide 2H-TaSe2. Contrary to an earlier conclusion from inelastic neutron scattering that found only partial softening of the phonon mode, here the authors demonstrate that the mode not only softens completely but even does so at a somewhat higher temperature than the CDW transition itself. They suggest that this indicates the presence of an additional precursor phase that exists in a narrow temperature range immediately above the CDW ordering transition.

First of all, the main result of this paper pertaining to the demonstration of a complete phonon softening is a very important finding. Indeed, 2H-TaSe2 is a textbook example of a CDW system, and the theory of a CDW transition implies an essential coupling between the CDW and the periodic lattice distortion. The condensation of CDW fluctuations into an ordered phase requires/implies that the lattice develops a static symmetry-breaking distortion, which in

accordance with the Goldstone theorem should be characterized by a gapless mode emanating from the ordering vector. The second-order character of the transition implies that this mode must soften continuously at temperatures above the transition. Therefore, the fact that this softening has not been observed in previous experiments is highly unexpected and cannot be reconciled with the well accepted theory of the CDW formation. In this respect, a clear demonstration of the existence of a fully softened mode is a long-anticipated and important result that, in my opinion, deserves the publication in Nature Communications.

This said, I find the second conclusion of the authors about the presence of a precursor phase above T_{CDW} much less convincing. This conclusion is essentially based on the extrapolation of data points in the T-dependence (Fig. 3a) to zero energy. These data points result, in turn, from the fits such as those in Fig. 2, where the peaks overlap strongly at temperatures below 130 K. Assuming that the intrinsic phonon signal in the vicinity of Q_{CDW} is given by a V-shaped phonon shape, something like $E = \sqrt{|\mathbf{q} - \mathbf{q}_0|^2 + \Delta^2}$, convoluting it with the experimental energy- and momentum resolution would result in a 3D scattering intensity distribution $I(\mathbf{q}, E)$. It is well known that the positions of maxima in the one-dimensional cuts of this function in energy or momentum would not coincide with the actual dispersion $E(\mathbf{q})$, and therefore extracting the true dispersion requires proper three-dimensional modeling of this function with full account of the resolution function, not just fits of one-dimensional cuts. It is very likely that by performing such one-dimensional analysis, the authors introduced a systematic error in their fitting results that increases towards T_{CDW} , resulting in a shift of the extrapolated point of phonon softening from its expected position. This analysis alone is therefore insufficient to claim the presence of another phase transition or an additional thermodynamic phase. Furthermore, no evidence for a transition at T^* has been observed in other probes such as specific heat.

If the authors insist on their conclusion about the presence of a precursor phase, I would ask them to demonstrate using 3D modeling of the intensity function obtained from a convolution of the expected phonon dispersion with the calculated instrumental resolution that their measured data are incompatible with the phonon softening happening at T_{CDW} . It would be also helpful to see the zoomed-in version of the part of Fig. 3(a) in the immediate vicinity of the transition and the corresponding dispersion curves (like those in Fig. 3c) for all temperatures between T_{CFW} and T^* . I find the step in q in Fig. 3(c) insufficiently detailed to see the U-shaped minimum in the gapped dispersion even for higher temperatures of 150 K and above, so it will be even less obvious at lower temperatures.

A minor technical comment: It would be helpful if the guides to the eye in Figs. 3(c),(d) show more physically realistic dependences. Why is the curve at 250 K approximated with a smooth curve, and those at 200 and 150 K are just interpolated linearly between the data points? For all curves with a finite energy gap, the dispersion cannot have a cusp at the minimum of the dispersion, therefore the current guides to the eye are actually misleading the eye. I suggest using some empirical model for the dispersion instead, so that all curves are treated in the same way.

In summary, I find this paper is potentially publishable in Nature Communications if the authors can refute the hypothesis that the deviation of T^* from T_{CDW} is an artifact of finite resolution effects and the particular fitting of the energy- and momentum-distribution curves.

Reviewer #3:

Remarks to the Author:

The manuscript reports a primarily high-resolution inelastic X-ray scattering (IXS) study of the charge density wave (CDW) formation in layered transition-metal dichalcogenide, 2H-TaSe₂. The work is a significant advancement compared to the previous inelastic neutron scattering studies on this material carried out in the 1970-s (Refs. 25 and 28). Specifically, by carefully analyzing their measured X-ray scattering profiles, authors are able to precisely follow the temperature dependence of both the energy and the lifetime of the phonon mode softening at the CDW transition. What they discover, is the pre-cursor short-range-ordered CDW phase with already softened overdamped phonon, which exists within several Kelvin temperature range ($\sim 123\text{K}$ to $\sim 129\text{K}$), above the temperature where long-range CDW is established ($\sim 122\text{K}$). The IXS data are

supplemented by the ARPES measurements, which correlate the observed lattice structural and dynamical behaviors with the electronic structure, indicating that they have only a very small, albeit measurable impact on electronic properties. Finally, a DFT analysis is presented showing potential importance of the low-energy soft phonon spectrum investigated here by IXS for the dramatic enhancement of superconductivity in this material at high pressure, the result that is mainly motivational but also with a broad appeal to diverse readership of Nature Communications. The work presents expertly performed measurements, which are also carefully and scrupulously analyzed and quite eloquently described. I think this is an interesting work with a broad appeal to the diverse scientific audience and recommend that it is accepted for publication in Nature Communications. Below, I list several minor technical comments, which authors should address in preparing their manuscript for publication.

Line 70. "The much better momentum resolution of IXS compared to INS ..." -> angular resolution only? I would think the longitudinal momentum resolution is coupled to energy resolution - perhaps something could be added in methods on this?

Line 129. "in reasonable agreement with the coherence length of 14 Å estimated from specific heat measurements." - coherence length of what? Please specify.

Line 150. "symmetrically related" -> symmetry-related

Line 198. " but strongly responds to" -> I would remove "strongly".

Line 202. "the impact of the CDW soft mode regard the strongly increasing superconducting transition temperature" -> "regard"?

Line 493-494. Figure 4 caption. "The red 494 solid line represents the same power law fit as shown in Fig. 2(c)" - Fig. 2(c) looks like a misnomer here.

Figure 4 - the break in the Y-axis in panel (e) is confusing. It is unclear to which part of the scale points to the left of the red dashed line including those marked (a), (b), (c) belong. I suggest splitting this panel into two different panels or using the left and right scales for "CDW peak" and "phonon" intensity.

Temperature dependence shown in Fig. 4 was measured in a set of energy scans at $q=(0.325,0,0)$, which authors call q_{CDW} , with high wave vector resolution of 0.018 1/\AA .

However, peak position in q moves in the temperature range where the T-dependence was measured, changing from 0.323 at $T_{\text{CDW}} = 122.3\text{K}$ to 0.333 in commensurate phase below 88K. The changing position of the peak would impact the measured T-dependence of the peak intensity. Perhaps, this effect is of little or no importance, and in addition it has been investigated by the authors by performing elastic wavevector scans reported in the next Figure. Nevertheless, authors should discuss the impact of the changing peak position on the systematic error of their intensity measurement in Fig. 4 (perhaps with reference to the follow-up Fig. 5) and incorporate it in the error bar.

Reviewer #4:

Remarks to the Author:

Xingchen Shen et al. employed Inelastic X-ray Scattering (IXS) and Angle Resolved Photoemission (ARPES) to study a phenomenon of charge density wave (CDW) transition in the single-crystalline sample of 2H-TaSe₂, a material where CDW has been first detected experimentally about 50 years ago. IXS delivered the data about phonons: dispersion relations at various temperatures above and below ~ 122 K, a temperature of the CDW transition reported in the previous studies; ARPES measurements yielded contours of the Fermi surface at 126 K. IXS (as well as the neutron scattering) and ARPES are in essence the common techniques used in the studies of the CDW materials. On that account the authors have followed the valid experimental methodology. The abstract states that the results of this study rule out the "central-peak" scenario of the CDW transition in 2H-TaSe₂ and provide evidence for a novel "precursor" phase developing above the transition temperature. The authors believe that the hallmark of the latter phase is the "complete" softening of the phonon mode. Given the substantial amount of reports dealing with the CDW transitions in the transition metal di-chalcogenides these two results could potentially warrant a publication in Nature Communications, a journal aiming at disseminating important advances. I should confess that I was not aware of the "central-peak" scenario of the CDW transition. I also

was not able to find any mentioning of it in the literature. While this (central) peak has been observed in the numerous studies of the structural phase transitions, it seems it has not yet been invoked for explanation of the CDW transition. In essence, its origin has been debated. Recent IXS study of YBCO (Nature Physics 10, 52) attributed energy-resolution-limited elastic peak to the presence of CDW domains above and below the transition temperature. The major result discussed by Xingchen Shen et al. seems to be based on the erroneous belief that "central peak scenario" has been considered and debated as a possible mechanism of the CDW transition in 2H-TaSe₂. The other observation, that is the softening of the phonon mode, is hardly new. It was studied in the seminal work by D. E. Moncton, J. D. Axe and F. J. DiSalvo (PRB 16, 801), albeit these authors admitted that insufficient resolution might have prevented the detailed analysis of the sharp anomaly in the dispersion (Figure 3a of the present manuscript supports this assumption). Nevertheless, the temperature dependence of the softening of the phonon was firmly established decades ago.

To conclude, the paper communicates results of the excellent IXS study of 2H-TaSe₂ but it neither conveys any new findings nor does it resolve a long standing controversy since I believe that "central-peak scenario" has never been discussed in connection with the mechanism of the CDW transition. Observation of the softening of the phonon mode is in line with the previous studies but it cannot dislodge a model which has never (in my opinion) existed.

I shall admit that "central peak scenario" might have been mentioned in the reports which escaped my attention. However, it is my opinion that the manuscript cannot be published in its present form.

Reviewer #1 (Remarks to the Author):

Report on the manuscript 'Precursor phase with full phonon softening above the CDW phase transition in 2H-TaSe2'

The work by X. Shen et al report a comprehensive lattice dynamics study of the 2D layered material TaSe2 by means of inelastic x-ray scattering and ARPES. The authors find, in contrast with the previous neutron data taken in the 70's, a complete phonon softening at a temperature higher than the CDW transition. The softening is characterised by a critically damped phonon with a critical exponent lower than the mean field value, highlighting the role of the high temperature charge fluctuations. A detailed discussion of the results in a broad context of TMDs and CDW systems is also included. Finally, the main conclusion is that the Fermi surface nesting scenario is the main driving force of the CDW, unlike other TMDs like VSe2 and NbSe2.

The paper is well written, the experiments were carefully performed and the data analysed. I believe that the manuscript deserves to be published in Nat. Comm. as it brings new information to understand the origin of TMDs and CDWs in general. However, I think that the papers can be improved:

- The authors say that the FS scenario seems to drive the phase transition, instead of EPI. This seems or does not seem to be the case. The softening of the phonon in momentum space is not so sharp as in ZrTe3, although not so broad as the NbSe2, but it still develops 'some broadening'. I ask the authors to quantify the FS, i.e. real and imaginary parts of the electronic susceptibility and momentum dependence EPI and check whether 'peaks' appear and how broad they are. I understand that these calculations can be rather straightforward (although computationally expensive) given the expertise of the group. These calculations will also give a push to the manuscript in Nat. Comm.

We thank the referee for the positive assessment of our work and provide more details about the underlying mechanisms driving the CDW transitions in our calculations. A detailed discussion is given in supplemental note 3 (1 page additional text) along with three new figures S3-S5. In our calculations, we cannot extract the real part of the electronic susceptibility and, therefore, provide an analysis of the electronic joint density of states (eJDOS) which is closely related to the imaginary part of the susceptibility, also sometimes referred to as the nesting function. Thus, it reflects directly the Fermi surface properties and is particularly interesting with regard to the prototypical Peierls model for CDW in a one-dimensional metallic chain.

- The phonon softens at $T > T_{CDW}$ (7-9 K above T_{CDW}), then, is the softening a precursor of the CDW phase at a different q ? or, competing? I do not find a proper discussion on the origin of the sharp increase of the elastic line at T_{CDW} . If not phonon driven, is this order-disorder, chemical bond instability type,...

The phonon softening at $T > T_{CDW}$ is a precursor of the CDW phase in our understanding. Combined with comments from another referee, we became aware that the use of "precursor phase" might not be correct since our results do not unambiguously determine the state of 2H-TaSe2 at $T_{CDW} \leq T \leq T^$ as a distinct thermodynamic phase. Therefore, we adopted the expression "precursor region" in our title and throughout the text.*

Furthermore, we have added a new paragraph in the discussion on this topic (2nd paragraph in section DISCUSSION). For a more detailed understanding model calculations would be important which are able to include temperature effects as well as can cope with fluctuations.

Finally, I would suggest the authors to improve the introduction and make it more Nature style. The intro is too narrow, focusing only in TaSe₂ and TMDs. I would suggest to broaden it: cuprates, kagomes,... Note that in the cuprates the stripe order seems to lock the high temperature charge fluctuations to the commensurate value. Also, in the Kagomes, the dynamics of CDW does not show a soft mode, but the elastic line increases sharply defining an order-disorder phase transition... With addressing these comments, I recommend the paper to be published in Nat Comm.

In the revised introduction we put more emphasis on the broader context of our investigation and refer to lattice dynamical studies in cuprates, transition-metal-based Kagome materials as well as other classic CDW systems.

Reviewer #2 (Remarks to the Author):

In the present paper, X. Shen et al. employ inelastic x-ray scattering to solve a decades-old puzzle about the soft phonon mode at the CDW ordering vector in the layered transition-metal chalcogenide 2H-TaSe₂. Contrary to an earlier conclusion from inelastic neutron scattering that found only partial softening of the phonon mode, here the authors demonstrate that the mode not only softens completely but even does so at a somewhat higher temperature than the CDW transition itself. They suggest that this indicates the presence of an additional precursor phase that exists in a narrow temperature range immediately above the CDW ordering transition.

First of all, the main result of this paper pertaining to the demonstration of a complete phonon softening is a very important finding. Indeed, 2H-TaSe₂ is a textbook example of a CDW system, and the theory of a CDW transition implies an essential coupling between the CDW and the periodic lattice distortion. The condensation of CDW fluctuations into an ordered phase requires/implies that the lattice develops a static symmetry-breaking distortion, which in accordance with the Goldstone theorem should be characterized by a gapless mode emanating from the ordering vector. The second-order character of the transition implies that this mode must soften continuously at temperatures above the transition. Therefore, the fact that this softening has not been observed in previous experiments is highly unexpected and cannot be reconciled with the well accepted theory of the CDW formation. In this respect, a clear demonstration of the existence of a fully softened mode is a long-anticipated and important result that, in my opinion, deserves the publication in Nature Communications.

We thank the referee for the positive appraisal of our work. We believe that our analysis of the origin of the CDW in 2H-TaSe₂ (added in the revised version) further emphasizes this model character of 2H-TaSe₂.

This said, I find the second conclusion of the authors about the presence of a precursor phase above T_{CDW} much less convincing. This conclusion is essentially based on the extrapolation of data points in the T-dependence (Fig. 3a) to zero energy. These data points result, in turn, from the fits such as those in Fig. 2, where the peaks overlap strongly at temperatures below 130 K. Assuming that the intrinsic phonon signal in the vicinity of Q_{CDW} is given by a V-shaped phonon shape, something like $E = \sqrt{(|q - q_0|^2 + \Delta^2)}$, convoluting it with the experimental energy- and momentum resolution would result in a 3D scattering intensity distribution $I(q, E)$. It is well known that the positions of maxima in the one-dimensional cuts of this function in energy or momentum would not coincide with the actual dispersion $E(q)$, and therefore extracting the true dispersion requires proper three-dimensional modeling of this function with full account of the resolution function, not just fits of one-dimensional cuts. It is very likely that by performing such one-dimensional analysis, the authors introduced a systematic error in their fitting results that increases towards T_{CDW} , resulting in a shift of the extrapolated point of phonon softening from its expected position. This analysis alone is therefore insufficient to claim the presence of another phase transition or an additional thermodynamic phase. Furthermore, no evidence for a transition at T^* has been observed in other probes such as specific heat.

We agree with the referee that resolution effects can be misleading. However, a broad/coarse resolution makes it difficult to observe sharp features such as the phonon softening in 2H-TaSe₂. Indeed, this was the problem of the early INS experiments¹. The fact that we clearly see the softening is already proof that we have a good resolution. Thus, measurements with coarse resolution tend to underestimate a softening, i.e., the deduced phonon energies are larger than in reality.

We exemplify this behavior with data on the soft mode in SrTiO₃ (see figure and discussion further down). The data shows that the finite resolution of our phonon measurements rather lead to an

overestimation of the strongly renormalized phonon mode. Therefore, the discrepancy between T^* and T_{CDW} cannot be explained by resolution effects. We also note that no deviations between the temperature deduced from phonon softening and the onset of long-range ordered CDW superlattice peaks have been observed in the same experimental setup for $2H-NbSe_2$ or some rare-earth tritellurides^{3,4}. Furthermore, a large number of publications based on IXS measurements adopts a similar type of analysis and did not yet find a similar discrepancy between the temperature of the phonon softening and TCDW indicating long-range order.

However, the referee is right that the use of the expression “precursor phase” is misleading. Our data show the phonon softening at T^* but the characteristics at $T_{CDW} \leq T \leq T^*$ are more that of a suppressed CDW transition which fully occurs only at T_{CDW} . Therefore, we change our phrasing and use the term “precursor region” in the title and throughout the text. We highlight the particularity of this region which has not been observed – to our knowledge – in any other CDW material.

If the authors insist on their conclusion about the presence of a precursor phase, I would ask them to demonstrate using 3D modeling of the intensity function obtained from a convolution of the expected phonon dispersion with the calculated instrumental resolution that their measured data are incompatible with the phonon softening happening at T_{CDW} .

While we do not insist on the presence of a separate phase between T_{CDW} and T^* we would like to make a comment on the resolution effects. Rather than convoluting the expected dispersion with the resolution ellipsoid, we can present comparable measurements of a soft phonon mode in $SrTiO_3$ which we performed recently on the same IXS instrument. The anomaly of the soft mode at $\mathbf{q}_{cubic} = (0.5, 0.5, 0.5)$ in $SrTiO_3$ is similarly sharp in momentum space⁵ and we can learn how resolution may affect the deduced phonon energies.

We explain in Supplemental Note 1 that we reduced the aperture of the round analyzer opening to improve the angular, i.e., momentum resolution and be able to follow the temperature dependence of the elastic peak in $2H-TaSe_2$.

For $SrTiO_3$, we performed measurements of the soft phonon mode with different apertures of the round analyzer opening (Figure on the right). The data shown in blue correspond to the typical setup which we also used for our phonon measurements in $2H-TaSe_2$. We see that the deduced phonon energy for the typical setup slightly overestimates the soft mode energy (see inset). This is understandable since the broader momentum resolution probes a region in which the soft mode already disperses upwards away from the superlattice peak position, i.e., $\mathbf{q} = (0.5, 0.5, 0.5)$.

Therefore, we argue that the very low energy of the CDW soft phonon mode in $2H-TaSe_2$ at $T = T^*$ is not an artefact due to the finite resolution of our measurement.

It would be also helpful to see the zoomed-in version of the part of Fig. 3(a) in the immediate vicinity of the transition and the corresponding dispersion curves (like those in Fig. 3c) for all temperatures between T_{CFW} and T^* . I find the step in q in Fig. 3(c) insufficiently detailed to see the U-shaped minimum in the gapped dispersion even for higher temperatures of 150 K and above, so it will be even less obvious at lower temperatures.

The zoom of Figure 3(a) is shown in Figure 4(d). For $T < T^$, we measured the dispersion only for $T = 125$ K. The results are now included in Figure 3(c) and (d). Due to beam time limitations, the data at $T = 125$ K were only taken at the same momenta as the dispersion lines at $T \geq 150$ K. The results are (within the error bar) identical with those obtained at the respective wavevectors at $T = 130$ K and indicate that the phonon does not change significantly in this temperature range.*

We agree with the referee that it would be good to have a finer momentum step size close to q_{CDW} . However, this would only make sense if we simultaneously would improve the momentum resolution, e.g., as we did for $SrTiO_3$ (see above). However, the data for $SrTiO_3$ show that the phonon intensity decreases dramatically and there is simply not enough beam time to perform extended phonon measurements in high momentum resolution mode.

A minor technical comment: It would be helpful if the guides to the eye in Figs. 3(c),(d) show more physically realistic dependences. Why is the curve at 250 K approximated with a smooth curve, and those at 200 and 150 K are just interpolated linearly between the data points? For all curves with a finite energy gap, the dispersion cannot have a cusp at the minimum of the dispersion, therefore the current guides to the eye are actually misleading the eye. I suggest using some empirical model for the dispersion instead, so that all curves are treated in the same way.

We thank the referee for pointing out this issue. Rather than adjusting a function to the observed points on the dispersion we decided to only show straight connecting lines. This way it is clear that the lines have no physical meaning and that the dispersion in between the measured wave vectors is not exactly known.

In summary, I find this paper is potentially publishable in Nature Communications if the authors can refute the hypothesis that the deviation of T^* from T_{CDW} is an artifact of finite resolution effects and the particular fitting of the energy- and momentum-distribution curves.

We believe that we show evidence that the softening happens indeed above T_{CDW} and hope that the referee supports publication of the revised manuscript.

Reviewer #3 (Remarks to the Author):

The manuscript reports a primarily high-resolution inelastic X-ray scattering (IXS) study of the charge density wave (CDW) formation in layered transition-metal dichalcogenide, 2H-TaSe₂. The work is a significant advancement compared to the previous inelastic neutron scattering studies on this material carried out in the 1970-s (Refs. 25 and 28). Specifically, by carefully analyzing their measured X-ray scattering profiles, authors are able to precisely follow the temperature dependence of both the energy and the lifetime of the phonon mode softening at the CDW transition. What they discover, is the pre-cursor short-range-ordered CDW phase with already softened overdamped phonon, which exists within several Kelvin temperature range (~123K to ~129K), above the temperature where long-range CDW is established (~122K). The IXS data are supplemented by the ARPES measurements, which correlate the observed lattice structural and dynamical behaviors with the electronic structure, indicating that they have only a very small, albeit measurable impact on electronic properties. Finally, a DFT analysis is presented showing potential importance of the low-energy soft phonon spectrum investigated here by IXS for the dramatic enhancement of superconductivity in this material at high pressure, the result that is mainly motivational but also with a broad appeal to diverse readership of Nature Communications.

The work presents expertly performed measurements, which are also carefully and scrupulously analyzed and quite eloquently described. I think this is an interesting work with a broad appeal to the diverse scientific audience and recommend that it is accepted for publication in Nature Communications. Below, I list several minor technical comments, which authors should address in preparing their manuscript for publication.

We thank the referee for his positive comments on our work and provide responses/revisions to his further comments in the following.

Line 70. "The much better momentum resolution of IXS compared to INS ..." -> angular resolution only? I would think the longitudinal momentum resolution is coupled to energy resolution - perhaps something could be added in methods on this?

We discuss this in the supplemental note 1.

Line 129. "in reasonable agreement with the coherence length of 14 Å estimated from specific heat measurements." - coherence length of what? Please specify.

It is the coherence length of the CDW order parameter. McMillan has shown that the observations for 2H-TaSe₂ can be understood in a phonon-entropy model, i.e., that phonons provide most of the entropy at the CDW phase transition. He introduces a coherence length ξ_0 which is related to the range in reciprocal space over which renormalization of the soft mode is expected. Craven et al. estimated $\xi_0 = 14 \text{ \AA}$ based on their specific heat measurements [40]. We have clarified the corresponding section in the results part.

Line 150. "symmetrically related" -> symmetry-related

Ok

Line 198. " but strongly responds to" -> I would remove "strongly".

Ok.

Line 202. "the impact of the CDW soft mode regard the strongly increasing superconducting transition temperature" -> "regard"?

We corrected the sentence: "role of the CDW soft phonon mode with regard to the increasing superconducting temperatures at high pressure".

Line 493-494. Figure 4 caption. "The red solid line represents the same power law fit as shown in Fig. 2(c)" - Fig. 2(c) looks like a misnomer here.

We corrected it. It refers now to Figure 3(a).

Figure 4 - the break in the Y-axis in panel (e) is confusing. It is unclear to which part of the scale points to the left of the red dashed line including those marked (a), (b), (c) belong. I suggest splitting this panel into two different panels or using the left and right scales for "CDW peak" and "phonon" intensity.

In order to avoid confusion we deleted the break and show all on one scale.

Temperature dependence shown in Fig. 4 was measured in a set of energy scans at $q=(0.325,0,0)$, which authors call q_{CDW} , with high wave vector resolution of 0.018 1/\AA . However, peak position in q moves in the temperature range where the T-dependence was measured, changing from 0.323 at $T_{\text{CDW}} = 122.3\text{K}$ to 0.333 in commensurate phase below 88K. The changing position of the peak would impact the measured T-dependence of the peak intensity. Perhaps, this effect is of little or no importance, and in addition it has been investigated by the authors by performing elastic wavevector scans reported in the next Figure. Nevertheless, authors should discuss the impact of the changing peak position on the systematic error of their intensity measurement in Fig. 4 (perhaps with reference to the follow-up Fig. 5) and incorporate it in the error bar.

All energy/phonon scans (including those analysed in Figure 4) were performed with the standard momentum resolution of 0.09 \AA^{-1} , which is much broader than the shift of the CDW peak position on cooling below T_{CDW} . Furthermore, energy/phonon scans analyzed in Figure 4 were done at $T \geq 122.5 \text{ K}$ ($< T_{\text{CDW}}$). Here, even high-resolution scans at zero energy transfer (see Fig. 5(c)) do not find a significant temperature dependent wave vector of the peak due to CDW fluctuation/soft mode.

Reviewer #4 (Remarks to the Author):

Xingchen Shen et al. employed Inelastic X-ray Scattering (IXS) and Angle Resolved Photoemission (ARPES) to study a phenomenon of charge density wave (CDW) transition in the single-crystalline sample of 2H-TaSe₂, a material where CDW has been first detected experimentally about 50 years ago. IXS delivered the data about phonons: dispersion relations at various temperatures above and below ~122 K, a temperature of the CDW transition reported in the previous studies; ARPES measurements yielded contours of the Fermi surface at 126 K. IXS (as well as the neutron scattering) and ARPES are in essence the common techniques used in the studies of the CDW materials. On that account the authors have followed the valid experimental methodology. The abstract states that the results of this study rule out the “central-peak” scenario of the CDW transition in 2H-TaSe₂ and provide evidence for a novel “precursor” phase developing above the transition temperature. The authors believe that the hallmark of the latter phase is the “complete” softening of the phonon mode. Given the substantial amount of reports dealing with the CDW transitions in the transition metal di-chalcogenides these two results could potentially warrant a publication in Nature Communications, a journal aiming at disseminating important advances. I should confess that I was not aware of the “central-peak” scenario of the CDW transition. I also was not able to find any mentioning of it in the literature. While this (central) peak has been observed in the numerous studies of the structural phase transitions, it seems it has not yet been invoked for explanation of the CDW transition. In essence, its origin has been debated. Recent IXS study of YBCO (Nature Physics 10, 52) attributed energy-resolution-limited elastic peak to the presence of CDW domains above and below the transition temperature. The major result discussed by Xingchen Shen et al. seems to be based on the erroneous belief that “central peak scenario” has been considered and debated as a possible mechanism of the CDW transition in 2H-TaSe₂. The other observation, that is the softening of the phonon mode, is hardly new. It was studied in the seminal work by D. E. Moncton, J. D. Axe and F. J. DiSalvo (PRB 16, 801), albeit these authors admitted that insufficient resolution might have prevented the detailed analysis of the sharp anomaly in the dispersion (Figure 3a of the present manuscript supports this assumption). Nevertheless, the temperature dependence of the softening of the phonon was firmly established decades ago. To conclude, the paper communicates results of the excellent IXS study of 2H-TaSe₂ but it neither conveys any new findings nor does it resolve a long standing controversy since I believe that “central-peak scenario” has never been discussed in connection with the mechanism of the CDW transition. Observation of the softening of the phonon mode is in line with the previous studies but it cannot dislodge a model which has never (in my opinion) existed. I shall admit that “central peak scenario” might have been mentioned in the reports which escaped my attention. However, it is my opinion that the manuscript cannot be published in its present form.

We thank the referee for his review of our work and the positive assessment of the quality of our IXS study. However, we disagree with his statement that “the temperature dependence of the softening of the phonon was firmly established decades ago”. It was established that there is a temperature dependence. But important points including whether the phonon fully softens and, if, at what temperature were not known. Certainly, a full phonon softening well above the CDW phase transition temperature has not been reported or discussed for 2H-TaSe₂, another transition-metal dichalcogenide or (to our knowledge) in any CDW material.

The authors of the original phonon study¹ say that they cannot deduce too much detail on the lattice dynamics because of their coarse momentum resolution. However, they are confident enough to show the integrated intensity of the peak at zero energy transfer which rises just below 150 K, i.e., well-above T_{CDW} . Even if the phonon would soften at T_{CDW} , this alleged rise of the elastic peak at high temperature would have to be addressed in order to understand the origin of the CDW order in 2H-TaSe₂. Thus, our study provides important insights in that it does not reproduce this behavior. The

elastic line only rises above the phonon intensity level at temperatures below that at which the soft mode is fully softened.

We argue that it is useful to discuss our results with respect to the observation of a central peak since the corresponding results are clearly shown in Ref. ¹ and similar observation have been recently reported not only in cuprates but also in other, more conventional CDW materials like ZrTe₃. We also emphasize that the issue of a central peak is an observation rather than a fully understood model scenario, e.g., like the Peierls scenario for CDW formation in a 1D metal. The origin of the central peak is still not fully understood, even in well-known central-peak compounds such as SrTiO₃. Scattering-type, i.e., XRD investigations of 2H-TaSe₂ focused on the intriguing change from an incommensurate to a commensurate CDW at temperatures below 100K⁶. Thus, our study highlights that new techniques should be applied to “old” materials. Important points can be clarified or previous results be corrected.

References:

- 1 Moncton, D. E., Axe, J. D. & DiSalvo, F. J. Neutron scattering study of the charge-density wave transitions in 2H-TaSe₂ and 2H-NbSe₂. *Physical Review B* **16**, 801-819 (1977).
- 2 Weber, F., Rosenkranz, S., Castellán, J. P., Osborn, R., Hott, R., Heid, R., Bohnen, K. P., Egami, T., Said, A. H. & Reznik, D. Extended Phonon Collapse and the Origin of the Charge-Density Wave in 2H-NbSe₂. *Physical Review Letters* **107**, 107403 (2011).
- 3 Maschek, M., Zocco, D. A., Rosenkranz, S., Heid, R., Said, A. H., Alatas, A., Walmsley, P., Fisher, I. R. & Weber, F. Competing soft phonon modes at the charge-density-wave transitions in DyTe₃. *Physical Review B* **98**, 094304, doi:10.1103/PhysRevB.98.094304 (2018).
- 4 Maschek, M., Rosenkranz, S., Heid, R., Said, A. H., Giraldo-Gallo, P., Fisher, I. R. & Weber, F. Wave-vector-dependent electron-phonon coupling and the charge-density-wave transition in TbTe₃. *Physical Review B* **91**, 235146 (2015).
- 5 Stirling, W. G. Neutron inelastic scattering study of the lattice dynamics of strontium titanate: harmonic models. *Journal of Physics C: Solid State Physics* **5**, 2711, doi:10.1088/0022-3719/5/19/005 (1972).
- 6 Leininger, P., Chernyshov, D., Bosak, A., Berger, H. & Inosov, D. S. Competing charge density waves and temperature-dependent nesting in 2H-TaSe₂. *Physical Review B* **83**, 233101 (2011).

Reviewers' Comments:

Reviewer #1:

Remarks to the Author:

The authors have successfully addressed all my comments thus I support the publication of the manuscript in Nat Comm.

Just minor corrections:

- line 270 2H-VSe₂->1T-VSe₂.

- I would warn the authors about the new ref. [6] on kagomes. The chirality of the CDW in AV₃Sb₅ is being largely questioned and the STM data was not reproducible by other groups.

Comprehensive description of the CDW in AV₃Sb₅, relevant for TaSe₂, can be found in Phys. Rev. X 11, 031050 (2021), Nature Communications 14, 1015 (2023), Phys. Rev. B 105, L140501 (2022), etc.

Reviewer #2:

Remarks to the Author:

In my opinion, the authors have only partly addressed my comments raised in the previous report.

In particular, their argumentation regarding the influence of experimental resolution on the extraction of phonon softening temperature is not satisfactory. They argue based on an analogy with another compound, SrTiO₃, that finite momentum resolution would lead to an increase in the fitted peak positions instead of a decrease. While I agree with that, the effect of the energy resolution would be opposite. Two peaks that appear at a finite distance in energy that is smaller than the energy resolution would appear as a single peak in an energy scan and would be fitted to zero phonon energy while the actual value is finite. All curves shown for the SrTiO₃ data in the authors' reply are very far from this limit, as the peaks are well resolved in the data. To properly extract the peak positions, one would need proper modeling of the experimental intensity that involves multidimensional convolution with the resolution function that considers both energy and momentum resolution. This is a rather straightforward analysis, and unless this is properly done I am not convinced that the CDW transition and the phonon softening transition are indeed different as suggested by the presented fits. Considering that the authors admitted that there is no thermodynamic phase transition at T*, I find it very likely that the "precursor region" just represents a region where the phonon is so low in energy that it cannot be resolved within the experimental (energy) resolution. If this is indeed the case, this region would be of extrinsic origin, therefore properly refuting this hypothesis with numerical modeling is essential, especially given the very scarce data at low energies.

In a peak such as the one shown in Fig. 4(a-c), a single peak is fitted with a sum of a sharp elastic peak and two somewhat broader phonon modes. If the width of the phonon peak is treated as a free fitting parameter, it would be strongly coupled with the position of the peak, resulting in much larger error bars than those presented in panel (d). This is because two sharper peaks at slightly inelastic positions give nearly the same peak shape as one broader peak centered at zero. Because the intrinsic width of the peak depends both on momentum and on energy resolution functions, it is important to mention how the width of the peak was determined and what parameters were kept free in the fit for the determination of error bars.

Another point that caught my attention in the authors' reply is that they use the eJDOS instead of the proper nesting function (imaginary part of the Lindhard susceptibility) to evaluate the nesting properties of the Fermi surface. There is an essential difference between the two, namely the joint density of states would give a divergency (or large peak) at Q=0 or at a Q vector connecting similarly shaped Fermi pockets of the same type (hole-like with hole-like or electron-like with electron-like), whereas the nesting responsible for an electronic instability can only occur between a hole- and electron-like pockets, which is correctly captured by the Lindhard susceptibility only. In this respect, using the eJDOS instead of the Lindhard susceptibility for estimating the nesting properties is inaccurate and should be avoided. I do not see any reason why the authors cannot calculate the nesting function instead, this is rather straightforward.

With this said, I still consider that this paper is of sufficient interest for publication in Nature Communications, but the technical questions raised by the referees need to be addressed properly.

Reviewer #3:

Remarks to the Author:

In the revised manuscript authors have thoroughly addressed comments and suggestions included in my previous report from the first review round. I think that revisions have improved manuscript significantly and recommend that it is accepted for publication. Below, I list several minor revisions which authors might address when preparing the final resubmission of their manuscript.

1. From Fig. 3(b) it appears that the phonon mode at Q_{CDW} is not just a soft phonon below 130 K, but an over-damped one. This is an important observation, which would be useful to emphasize in the manuscript. It allows to rationalize why there is this broad precursor range of temperatures where lattice dynamics is soft but structural transition does not occur - an overdamped oscillator bears no dynamics (strictly speaking, its frequency is not zero, but is undefined). In other words, liquid-like, diffusive lattice dynamics, is present within the precursor region with softened but overdamped phonon at Q_{CDW} before it is overcome by truly zero-frequency (zero-energy) static order at T_{CDW} .

2. Line 158, "line in Fig. 4(e), note the break of the vertical scale" - now there is break in Fig. 4e, so this needs be removed/revised?

I must note that cannot concur with the opinion of Reviewer #4 that ruling out a central peak scenario for the CDW transition in 2H-TaSe is not important, or, as Reviewer #4 writes, "neither conveys any new findings nor does it resolve a long-standing controversy". The argument of that Reviewer that "I believe that "central peak scenario" has never been discussed in connection with the mechanism of the CDW transition" is not well founded. As reviewer themselves notes, the central peak phenomenon, which is observed in a rather broad range of phase transition, is an enigmatic phenomenon whose origin is still unclear and broadly debated. It is therefore natural to ask in the spirit of scientific inquiry, whether CDW is also possibly associated with central peak formation? Hence, I believe that ruling out the formation of a central peak without full phonon softening is an extremely important and timely finding reported in the manuscript, which provides insight both into the CDW formation mechanisms and the central peak enigma itself.

Reviewer #4:

Remarks to the Author:

In the revised manuscript Xingchen Shen et al have downplayed claim that their data rule out "central peak scenario" of the CDW transition in 2HTaSe2. In my first report I doubted that such a "scenario" was ever put forth while explaining the CDW transition in chalcogenides. Observation of the soft phonon mode at temperatures slightly exceeding accepted transition temperature ($\sim 122\text{K}$) seems to be the main result which potentially can warrant a report in Nature Communications. The upper boundary of the "precursor" region, that is 128.7 K, was found by fitting the experimental data plotted in Figure 3a (phonon energy versus temperature). 128.7K is just $\sim 7\text{K}$ over the commonly used transition temperature of $\sim 122\text{K}$. Unfortunately, the authors do not discuss quality of their samples and do not mention any transport measurements conducted on the same sample which was studied by the IXS. In my opinion such data should be included. As it happens, transition temperatures reported in the literature vary by $\sim 1\text{K}$. Moreover, recent study of TaS2 and TaSe2 detected increase of the transition temperatures in the "aged" samples (Appl. Phys. Lett. 118, 193105 (2021)). I believe it is imperative to present transport data indicating transition temperature when discussing precursor zone which only extends up to $\sim 7\text{K}$ above the CDW transition. The readers may also wonder about precision of the temperature measurements achieved in this work. Can the authors elaborate on how temperature was measured and calibrated? Is it possible that "precursor" region lies within the arrow bars of the measurement? Relevant information can be simply added to the Methods section of the paper.

Lastly, I like to comment on the inclusion of the ab-initio calculations and ARPES data. The DFPT results are interesting, but they mostly relate to the superconducting transition and enhancement of the critical temperature at high pressure. Both are valid subjects, but they seem to be not fitting well into the paper focusing on the "precursor" zone. Similarly, presented ARPES data are not

really discussed in the context of the main subject of the study. The figures display Fermi contours and spectra taken at a few points in the zone above and below the transition. The Fermi surfaces and observed shifts of the leading edges are in general agreement with the data reported in the literature. The authors do not discuss nesting features of the Fermi surfaces (which by the way are very difficult to read; reader not familiar with the previous ARPES studies of TaSe₂ will not be able to trace the contours). A nice example of such a discussion in connection to the CDW transition in di-chalcogenides could be found in *New Journal of Physics* 10, 125027 (2008). In order to put ARPES measurements in the context of the paper the authors of the present manuscript could have attempted extraction of the electron phonon coupling parameters from the ARPES data. In its present form the paper stands as a collection of independent studies (IXS, ARPES, DFT) loosely linked together by the sample. I have no doubts about quality of the data. But I doubt that the paper delivers any new findings which warrant publication in Nature journals. The manuscript would make an excellent publication in the specialized journals as it reports how improved momentum and energy resolution achievable in the modern IXS setups help to refine a picture of the phonon softening in a CDW material which was first drawn about 50 years ago (PRB 16, 801). Since I only doubt degree of the novelty, it should be left to the editors of Nature to decide whether this report constitutes a significant advance which may interest the broad audience. That said, I like to stress that without addition of the transport data clearly demonstrating a CDW transition at exactly 121.3K and detailed discussion covering precision of the temperature measurements existence of the precursor zone might be questioned by the initiated readers.

REVIEWER COMMENTS

Reviewer #1 (Remarks to the Author):

The authors have successfully addressed all my comments thus I support the publication of the manuscript in Nat Comm.

We thank the referee for the comments and for helping us to improve our manuscript.

Just minor corrections:

- line 270 2H-VSe₂->1T-VSe₂.

We have corrected the error.

- I would warn the authors about the new ref. [6] on kagomes. The chirality of the CDW in AV₃Sb₅ is being largely questioned and the STM data was not reproducible by other groups. Comprehensive description of the CDW in AV₃Sb₅, relevant for TaSe₂, can be found in Phys. Rev. X 11, 031050 (2021), Nature Communications 14, 1015 (2023), Phys. Rev. B 105, L140501 (2022), etc.

We thank the referee for pointing out this issue. We have included more references on this subject so that the reader can get a balanced impression from the given references.

Reviewer #2 (Remarks to the Author):

In my opinion, the authors have only partly addressed my comments raised in the previous report.

In particular, their argumentation regarding the influence of experimental resolution on the extraction of phonon softening temperature is not satisfactory. They argue based on an analogy with another compound, SrTiO₃, that finite momentum resolution would lead to an increase in the fitted peak positions instead of a decrease. While I agree with that, the effect of the energy resolution would be opposite. Two peaks that appear at a finite distance in energy that is smaller than the energy resolution would appear as a single peak in an energy scan and would be fitted to zero phonon energy while the actual value is finite. All curves shown for the SrTiO₃ data in the authors' reply are very far from this limit, as the peaks are well resolved in the data. To properly extract the peak positions, one would need proper modeling of the experimental intensity that involves multidimensional convolution with the resolution function that considers both energy and momentum resolution. This is a rather straightforward analysis, and unless this is properly done I am not convinced that the CDW transition and the phonon softening transition are indeed different as suggested by the presented fits. Considering that the authors admitted that there is no thermodynamic phase transition at T*, I find it very likely that the "precursor region" just represents a region where the phonon is so low in energy that it cannot be resolved within the experimental (energy) resolution. If this is indeed the case, this region would be of extrinsic origin, therefore properly refuting this hypothesis with numerical modeling is essential, especially given the very scarce data at low energies. In a peak such as the one shown in Fig. 4(a-c), a single peak is fitted with a sum of a sharp elastic peak and two somewhat broader phonon modes. If the width of the phonon peak is treated as a free fitting parameter, it would be strongly coupled with the position of the peak, resulting in much larger error bars than those presented in panel (d). This is because two sharper peaks at slightly inelastic positions give nearly the same peak shape as one broader peak centered at zero. Because the intrinsic width of the peak depends both on momentum and on energy resolution functions, it is important to mention how the width of the peak was determined and what parameters were kept free in the fit for the determination of error bars.

Indeed, the data set for SrTiO₃ primarily demonstrated how the finite momentum resolution affects phonon measurements in inelastic x-ray scattering.

The finite energy resolution is taken into account by using a damped harmonic oscillator function convolved with the energy resolution, in our case this is a pseudo-Voigt function as given in the Methods section (FWHM = 1.5 meV, Lorentzian contribution = 0.4). The ability to determine the phonon properties with high precision is due to the properties of the DHO function:

- (1) It simultaneously approximates the **two phonon peaks** at the Stokes (positive energy transfer) and anti-Stokes (negative energy transfer) sides of the spectrum with a **single set of parameters** for the intensity, the unrenormalized phonon energy $\tilde{\omega}_q$ and the damping Γ , where the renormalized phonon energy is defined as $\omega_q = \sqrt{\tilde{\omega}_q^2 - \Gamma^2}$.*
- (2) The intensity ratio of the peaks at the Stokes (positive energy transfer) and anti-Stokes (negative energy transfer) sides of the spectrum is determined by the **principle of detailed balance**. For example, see the data in Figure 2. The intensity ratio of the same phonon*

excitation at energy gain and loss was not a free parameter, but fixed by detailed balance for the given sample temperature.

- (3) For phonons with a large relative damping $\Gamma/\tilde{\omega}_q$, the energy dependence of the phonon intensity is important, which is expressed as a factor (phonon energy)⁻¹ in the single phonon scattering cross section. This is also taken into account by the DHO function we are using¹.

We illustrate the behavior of our fit for the case of the soft phonon mode in 2H-TaSe₂ at $T = 135$ K in **Figure D1**. We chose this data set because the raw data already have a single peak at $E = 0$, but our analysis is still able to determine a finite phonon energy of $\omega_q = \sqrt{\tilde{\omega}_q^2 - \Gamma^2} = 1.72$ meV, indicated by the small vertical blue bars.

It is instructive to plot the DHO function with the same parameters, but with near perfect resolution, i.e., close to zero (orange line in Fig. D1). We find that the experimental resolution accounts for only a very small fraction of the total broadening of the data. Rather, the broadening of the phonon due to the EPC dominates the shape of the scattering intensities and can be approximated with high accuracy. The fits of the soft phonon mode, as shown in Fig. D1 and in the manuscript, have only three free parameters, $\tilde{\omega}_q$, Γ and the phonon intensity. In effect, the analysis shown in Figure D1 (and similar data sets such as Figs. 2e, 4a-c) is a two-peak fit (DHO & elastic) to the main part of the data, -5 meV $\leq E \leq +5$ meV, with a total of four independent parameters (3 for the DHO, 1 for the elastic line [amplitude] since the shape of the elastic line is resolution limited).

The high statistics of the data allow the good error bars shown in our results. We explicitly point this out in supplemental note 2.

Thus, the energy convolution clearly shows full softening within the given experimental error bars. As discussed and shown in our previous response, effects of momentum resolution would only give an even smaller phonon energy for the soft mode, as any contribution from slightly different momentum transfer would be at a higher energy. Thus, the soft mode energy obtained with our fit is in fact an upper bound, and we therefore respectfully disagree with the referee that a multi-dimensional convolution is necessary to support our conclusion.

The temperature dependence of the soft mode shows a clear critical behavior with a critical temperature $T^* = 128.7$ K [Fig. 4(d)], while the superlattice peak intensity (measured on the same

Fig. D1. Data for 2H-TaSe₂ taken at $Q = (2.675, 0, 1)$ and $T = 135$ K. Dashed (red) line is a fit consisting of DHO functions convoluted with the experimental resolution (blue solid lines), estimated background (straight dotted line) and a resolution limited pseudo-Voigt function for the elastic line (black dash-dotted line). The green dashed line represents a resolution-limited DHO function, i.e., the damping Γ was set to zero, with a peak position corresponding to the renormalized frequency $\omega_q = 1.72$ meV deduced from the proper DHO fit (blue solid line, values for $\tilde{\omega}_q$ and Γ are given in the figure). The area of the resolution-limited DHO function was scaled to the area of the approximated one for the soft phonon mode.

sample with the same setup) shows a critical behavior with $T_c = 121.3$ K [see Fig. 5(b)(d)]. These are clearly outside any experimental uncertainties and show the existence of a precursor region.

Another point that caught my attention in the authors' reply is that they use the eJDOS instead of the proper nesting function (imaginary part of the Lindhard susceptibility) to evaluate the nesting properties of the Fermi surface. There is an essential difference between the two, namely the joint density of states would give a divergency (or large peak) at $Q=0$ or at a Q vector connecting similarly shaped Fermi pockets of the same type (hole-like with hole-like or electron-like with electron-like), whereas the nesting responsible for an electronic instability can only occur between a hole- and electron-like pockets, which is correctly captured by the Lindhard susceptibility only. In this respect, using the eJDOS instead of the Lindhard susceptibility for estimating the nesting properties is inaccurate and should be avoided. I do not see any reason why the authors cannot calculate the nesting function instead, this is rather straightforward.

We do not fully understand this comment of the referee.

In our work, we discuss the electronic joint density of states in comparison to the imaginary part of the susceptibility in the zero-energy and constant matrix element approximation, as given in the paper of Johannes and Mazin⁸

$$\lim_{\omega \rightarrow 0} \chi''(\mathbf{q}, \omega) / \omega = \sum_k \delta(\epsilon_{\vec{k}} - \epsilon_F) \delta(\epsilon_{\vec{k}+\vec{q}} - \epsilon_F),$$

where the authors write: "Since $\chi''(\mathbf{q}, \omega \rightarrow 0)$ is easier to calculate, it is often presented in first-principles studies as a quantitative test of the FS nesting (which it is) and ...". Except for the factor ω , the eJDOS is the same as the imaginary part of the susceptibility given by Johannes and Mazin. The results for the imaginary part of the susceptibility for 2H-TaSe2 as published by Johannes and Mazin⁸ are shown in Figure D2 in comparison to the eJDOS presented by us in Figure S4(b). However, we have changed the layout of the graph in Figure D2 to be similar to the published one. The results from Johannes and Mazin include the large increase at the Γ points and agree with our calculations in that the main nesting feature at finite wave vectors is located at $\mathbf{q} = (1/3, 1/3, 0)$ (see Fig. D2).

If the referee is referring to calculations going beyond the constant matrix element approximation, we note that this is beyond the current capabilities of our code. However, the approximations described above are widely used and accepted for testing the tendency towards Fermi surface nesting.

Imaginary part of the susceptibility of 2H-TaSe2 as shown in Fig. 4a in Johannes and Mazin, Physical Review B 77, 165135 (2008)

Fig. D2. (Top) Calculated imaginary part of the susceptibility published in [8]. (Bottom) Calculated eJDOS, also shown in Fig. S4(b), but displayed in a way similar to the data shown above.

In fact, our calculations shown in Fig. S5 and S6 go beyond the work of Johannes and Mazin. We demonstrate that the main nesting feature at $\mathbf{q} = (1/3, 1/3, 0)$ (see Fig. D2) is of minor relevance for the CDW in 2H-TaSe2 because it does not involve electron-phonon coupling. As explained in supplemental note 4, the nesting at $\mathbf{q} = (1/3, 1/3, 0)$ is related to intra-band scattering of one band at the Fermi level, whereas the EPC of the CDW soft phonon mode originates from inter-band scattering between the two bands forming the Fermi surface in 2H-TaSe2. The latter inter-band transition exhibits a maximum in the eJDOS at the CDW ordering wave vector, thus rationalizing the impact of the Fermi surface geometry on the formation of the CDW order. In this sense, nesting – but only of a particular inter-band electronic transition – plays an important role.

In fact, this scenario could be relevant for a number of materials, but has been largely overlooked in previous work. Recently, an orbital-selective CDW was also reported for TaTe4.⁹ In another study on the superconductor YNi2B2C,¹⁰ we also reported a strong electron-momentum dependence of the electron-phonon coupling.

With this said, I still consider that this paper is of sufficient interest for publication in Nature Communications, but the technical questions raised by the referees need to be addressed properly. *We thank the referee for emphasizing the general interest of our work, and we trust that we have shown that our results are robust and that there are no further technical issues to be addressed.*

Reviewer #3 (Remarks to the Author):

In the revised manuscript authors have thoroughly addressed comments and suggestions included in my previous report from the first review round. I think that revisions have improved manuscript significantly and recommend that it is accepted for publication. Below, I list several minor revisions which authors might address when preparing the final resubmission of their manuscript.

We thank the referee for the comments and suggestions that helped us improve our manuscript.

1. From Fig. 3(b) it appears that the phonon mode at Q_{CDW} is not just a soft phonon below 130 K, but an over-damped one. This is an important observation, which would be useful to emphasize in the manuscript. It allows to rationalize why there is this broad precursor range of temperatures where lattice dynamics is soft but structural transition does not occur - an overdamped oscillator bears no dynamics (strictly speaking, its frequency is not zero, but is undefined). In other words, liquid-like, diffusive lattice dynamics, is present within the precursor region with softened but overdamped phonon at Q_{CDW} before it is overcome by truly zero-frequency (zero-energy) static order at T_{CDW} .

We thank the referee for pointing out this particular aspect. We now use the terms “full phonon softening” and “overdamped phonon” to emphasize this type of behavior. We have also added a sentence to the discussion of recent work on the connection between phonon softening/damping and diffusion:

“The presence of an overdamped phonon over a finite temperature range bears similarities to a liquid-like scenario for the soft phonon mode^{11,12}. For instance, Ag-based argyrodites, which are promising candidates for thermoelectric applications, feature a structural phase transition temperature T_s at which a phonon softens on heating towards T_s . This soft optical phonon mode becomes and stays overdamped at temperatures $T \geq T_s$ and is linked to diffusion of Ag ions in these materials^{11,13}.”

2. Line 158, "line in Fig. 4(e), note the break of the vertical scale" - now there is break in Fig. 4e, so this needs be removed/revised?

We corrected the error.

I must note that cannot concur with the opinion of Reviewer #4 that ruling out a central peak scenario for the CDW transition in 2H-TaSe is not important, or, as Reviewer #4 writes, "neither conveys any new findings nor does it resolve a long-standing controversy". The argument of that Reviewer that "I believe that “central peak scenario” has never been discussed in connection with the mechanism of the CDW transition" is not well founded. As reviewer themselves notes, the central peak phenomenon, which is observed in a rather broad range of phase transition, is an enigmatic phenomenon whose origin is still unclear and broadly debated. It is therefore natural to ask in the spirit of scientific inquiry, whether CDW is also possibly associated with central peak formation? Hence, I believe that ruling out the formation of a central peak without full phonon softening is an extremely important and timely finding reported in the manuscript, which provides insight both into the CDW formation mechanisms and the central peak enigma itself.

We thank the referee for stating his opinion on this matter – which we fully agree with.

Reviewer #4 (Remarks to the Author):

In the revised manuscript Xingchen Shen et al have downplayed claim that their data rule out “central peak scenario” of the CDW transition in 2HTaSe2. In my first report I doubted that such a “scenario” was ever put forth while explaining the CDW transition in chalcogenides. Observation of the soft phonon mode at temperatures slightly exceeding accepted transition temperature ($\sim 122\text{K}$) seems to be the main result which potentially can warrant a report in Nature Communications. The upper boundary of the “precursor” region, that is 128.7K , was found by fitting the experimental data plotted in Figure 3a (phonon energy versus temperature). 128.7K is just $\sim 7\text{K}$ over the commonly used transition temperature of $\sim 122\text{K}$. Unfortunately, the authors do not discuss quality of their samples and do not mention any transport measurements conducted on the same sample which was studied by the IXS. In my opinion such data should be included. As it happens, transition temperatures reported in the literature vary by $\sim 1\text{K}$. Moreover, recent study of TaS2 and TaSe2 detected increase of the transition temperatures in the “aged” samples (Appl. Phys. Lett. 118, 193105 (2021)). I believe it is imperative to present transport data indicating transition temperature when discussing precursor zone which only extends up to $\sim 7\text{K}$ above the CDW transition. The readers may also wonder about precision of the temperature measurements achieved in this work. Can the authors elaborate on how temperature was measured and calibrated? Is it possible that “precursor” region lies within the arrow bars of the measurement? Relevant information can be simply added to the Methods section of the paper.

We note that in our experiment we determined both T^ [phonon softening, Fig. 4(d)] and T_{CDW} [XRD superlattice peak intensity, Fig. 5(b), and linewidth, Fig. 5(d)] independently. The x-ray measurements from which we determined both temperatures were done in one experimental session. Thus, we measured the same sample with the same thermometer under the same environmental conditions, including the way the sample was attached to the cold finger of the closed-cycle refrigerator. Temperatures in this setup are reproducible to within $\pm 0.1\text{K}$. Thus, the difference $T^* - T_{\text{CDW}} = 7.4\text{K}$ is much larger than the experimental uncertainty.*

We have added the new Figure S1 and supplemental note 1, which provide details on resistivity measurements on a sample from the same growth batch. Details on the used thermometers (calibrated cernox sensor) and how they were connected to the sample are also given in the notes. The agreement between the CDW transition temperatures $T_{\text{CDW, resist.}} = 121.7(5)\text{K}$ (see Fig. S1) and $T_{\text{CDW, x-ray}} = 121.3(2)\text{K}$ is good. Note that the latter is the result of a fit as shown in Figure 5(b)(d). The experimental data as shown in Figure 5(b)(d) itself demonstrates without any fitting that the transition is below 122K and above 121K . The good agreement between the transition temperature $T_{\text{CDW, resist.}}$ and $T_{\text{CDW, x-ray}}$ for the onset of the long-range charge-ordered phase confirms the significant difference to $T^ = 128.7\text{K}$, where the phonon becomes overdamped on cooling.*

Lastly, I like to comment on the inclusion of the ab-initio calculations and ARPES data. The DFPT results are interesting, but they mostly relate to the superconducting transition and enhancement of the critical temperature at high pressure. Both are valid subjects, but they seem to be not fitting well into the paper focusing on the “precursor” zone. Similarly, presented ARPES data are not really discussed in the context of the main subject of the study. The figures display Fermi contours and spectra taken at a few points in the zone above and below the transition. The Fermi surfaces and observed shifts of the leading edges are in general agreement with the data reported in the literature. The authors do

not discuss nesting features of the Fermi surfaces (which by the way are very difficult to read; reader not familiar with the previous ARPES studies of TaSe₂ will not be able to trace the contours). A nice example of such a discussion in connection to the CDW transition in di-chalcogenides could be found in New Journal of Physics 10, 125027 (2008). In order to put ARPES measurements in the context of the paper the authors of the present manuscript could have attempted extraction of the electron phonon coupling parameters from the ARPES data.

The goal of our ARPES measurements was to test the previously used quantities, such as the pseudogap as defined in Ref. ¹⁴ in our own samples and to see if we could find any anomaly upon crossing T. We emphasize this particular goal in our text (main and SI) and refer to the more detailed reports on the electronic band structure published previously. We are aware of the previous detailed reports on ARPES experiments on 2H-TaSe₂, which we cite in our manuscript and supplementary information, including the reference New Journal of Physics 10, 125027 (2008).*

In its present form the paper stands as a collection of independent studies (IXS, ARPES, DFT) loosely linked together by the sample.

We respectfully disagree with this statement and believe that our systematic multi-technique approach is essential for a microscopic understanding of the transition behavior. As mentioned above, there is a clear motivation for the ARPES measurements in our study, i.e., to verify the pseudogap behavior in our own samples. While this is may not be done often, it excludes the possibility that our samples behave differently for probes used in previous publications.

The calculations are closely linked to our x-ray measurements, where the referee points out the only part that we have included as a perspective for future experimental investigations [superconductivity under pressure, Fig. 6(d)]. The results for ambient pressure [black lines in Figure 6(a)-(c)] rationalize the phonon instability in 2H-TaSe₂ [Fig. 6(a)], demonstrate the agreement between the calculated and measured phonon linewidths of the CDW soft mode [Fig. 6(b)], and relate it to the electronic band structure [Fig. 6(c)]. Further details in the SI [Fig. S4-S6] go well beyond previous reports⁸ in that we explain why the main nesting feature at $q = (1/3, 1/3, 0)$ is irrelevant for CDW formation and that only a particular transition between two different electronic bands carries significant electron-phonon coupling strength.

I have no doubts about quality of the data. But I doubt that the paper delivers any new findings which warrant publication in Nature journals. The manuscript would make an excellent publication in the specialized journals as it reports how improved momentum and energy resolution achievable in the modern IXS setups help to refine a picture of the phonon softening in a CDW material which was first drawn about 50 years ago (PRB 16, 801). Since I only doubt degree of the novelty, it should be left to the editors of Nature to decide whether this report constitutes a significant advance which may interest the broad audience. That said, I like to stress that without addition of the transport data clearly demonstrating a CDW transition at exactly 121.3K and detailed discussion covering precision of the temperature measurements existence of the precursor zone might be questioned by the initiated readers.

We thank the reviewer for acknowledging the high quality of our data. We have added the requested resistivity data and highlighted the link between IXS, DFPT and ARPES in our manuscript, and we believe in the novelty and importance of our results, along the assessment given by Reviewer #3.

References:

- 1 Fåk, B. & Dorner, B. Phonon line shapes and excitation energies. *Physica B: Condensed Matter* **234-236**, 1107-1108 (1997).
- 2 Weber, F., Rosenkranz, S., Castellan, J. P., Osborn, R., Hott, R., Heid, R., Bohnen, K. P., Egami, T., Said, A. H. & Reznik, D. Extended Phonon Collapse and the Origin of the Charge-Density Wave in 2H-NbSe₂. *Physical Review Letters* **107**, 107403 (2011).
- 3 Maschek, M., Rosenkranz, S., Heid, R., Said, A. H., Giraldo-Gallo, P., Fisher, I. R. & Weber, F. Wave-vector-dependent electron-phonon coupling and the charge-density-wave transition in TbTe₃. *Physical Review B* **91**, 235146 (2015).
- 4 Maschek, M., Zocco, D. A., Rosenkranz, S., Heid, R., Said, A. H., Alatas, A., Walmsley, P., Fisher, I. R. & Weber, F. Competing soft phonon modes at the charge-density-wave transitions in DyTe₃. *Physical Review B* **98**, 094304, doi:10.1103/PhysRevB.98.094304 (2018).
- 5 Merritt, A. M., Weber, F., Castellan, J. P., Wolf, T., Ishikawa, D., Said, A. H., Alatas, A., Fernandes, R. M., Baron, A. Q. R. & Reznik, D. Nematic Correlation Length in Iron-Based Superconductors Probed by Inelastic X-Ray Scattering. *Physical Review Letters* **124**, 157001, doi:10.1103/PhysRevLett.124.157001 (2020).
- 6 Hoesch, M., Bosak, A., Chernyshov, D., Berger, H. & Krisch, M. Giant Kohn Anomaly and the Phase Transition in Charge Density Wave ZrTe₃. *Physical Review Letters* **102**, 086402 (2009).
- 7 Souliou, S. M., Lacmann, T., Heid, R., Meingast, C., Frachet, M., Paolasini, L., Haghighirad, A. A., Merz, M., Bosak, A. & Le Tacon, M. Soft-Phonon and Charge-Density-Wave Formation in Nematic BaNi₂As₂. *Physical Review Letters* **129**, 247602, doi:10.1103/PhysRevLett.129.247602 (2022).
- 8 Johannes, M. D. & Mazin, I. I. Fermi surface nesting and the origin of charge density waves in metals. *Physical Review B* **77**, 165135 (2008).
- 9 Xu, R. Z., Du, X., Zhou, J. S., Gu, X., Zhang, Q. Q., Li, Y. D., Zhao, W. X., Zheng, F. W., Arita, M., Shimada, K., Kim, T. K., Cacho, C., Guo, Y. F., Liu, Z. K., Chen, Y. L. & Yang, L. X. Orbital-selective charge-density wave in TaTe₄. *npj Quantum Materials* **8**, 44, doi:10.1038/s41535-023-00573-8 (2023).
- 10 Kurzhals, P., Kremer, G., Jaouen, T., Nicholson, C. W., Heid, R., Nagel, P., Castellan, J.-P., Ivanov, A., Muntwiler, M., Rumo, M., Salzmann, B., Strocov, V. N., Reznik, D., Monney, C. & Weber, F. Electron-momentum dependence of electron-phonon coupling underlies dramatic phonon renormalization in YNi₂B₂C. *Nat Commun* **13**, 228, doi:10.1038/s41467-021-27843-y (2022).
- 11 Ren, Q., Gupta, M. K., Jin, M., Ding, J., Wu, J., Chen, Z., Lin, S., Fabelo, O., Rodríguez-Velamazán, J. A., Kofu, M., Nakajima, K., Wolf, M., Zhu, F., Wang, J., Cheng, Z., Wang, G., Tong, X., Pei, Y., Delaire, O. & Ma, J. Extreme phonon anharmonicity underpins superionic diffusion and ultralow thermal conductivity in argyrodite Ag₈SnSe₆. *Nature Materials*, doi:10.1038/s41563-023-01560-x (2023).
- 12 Liu, H., Shi, X., Xu, F., Zhang, L., Zhang, W., Chen, L., Li, Q., Uher, C., Day, T. & Snyder, G. J. Copper ion liquid-like thermoelectrics. *Nature Materials* **11**, 422-425, doi:10.1038/nmat3273 (2012).
- 13 Shen, X., Koza, M. M., Tung, Y.-H., Ouyang, N., Yang, C.-C., Wang, C., Chen, Y., Willa, K., Heid, R., Zhou, X. & Weber, F. Soft Phonon Mode Triggering Fast Ag Diffusion in Superionic Argyrodite Ag₈GeSe₆. *Small n/a*, 2305048, doi:<https://doi.org/10.1002/sml.202305048> (2023).
- 14 Borisenko, S., Kordyuk, A., Yaresko, A., Zabolotnyy, V., Inosov, D., Schuster, R., Büchner, B., Weber, R., Follath, R., Patthey, L. & Berger, H. Pseudogap and Charge Density Waves in Two Dimensions. *Physical Review Letters* **100**, 196402 (2008).

Reviewers' Comments:

Reviewer #2:

Remarks to the Author:

I thank the authors for the detailed response, which I find mostly convincing. Any further discussion would go into technical details that are probably not so important for the publication decision, so I tend to recommend the paper for publication in its current version. Below I just want to respond to the authors' replies to my comments.

1. Regarding the discussion of resolution effects, one has to understand that the assumption that the spectrum can be fitted by two symmetric peaks corresponding to Stokes and anti-Stokes processes with damping is an approximation. In reality, when considering both momentum and energy resolutions, the peak intensity will have contributions from an infinite number of phonons in the vicinity of the q vector. At the point where the dispersion relationship is nonlinear (which is the case at $E=0$), this can lead to an asymmetric shape of the peaks and the deviation of the fitted peak positions from the true peak positions. With this said, I admit that the one-dimensional type of analysis employed by the authors is considered standard in the field.

2. Regarding the calculation of the nesting vectors, the paper by Johannes and Mazin emphasizes the nesting occurs at wave vectors where the `_real_` part of the Lindhard function has its maxima. This is consistent with the criterion for the CDW instability, as described for example in S.-K. Chan and V. Heine, *J. Phys. F: Met. Phys.* 3 795 (1973). The quantity relevant for the formation of the CDW is χ_0 in the static limit. This has no contribution from the imaginary part which vanishes at $E=0$ as an odd function of energy. The confusion therefore seems to arise because different authors use the word "nesting" in different senses. If one means just overlapping Fermi surfaces (joint density of states), then the imaginary part of χ_0 must be considered, but this is unrelated to the CDW instability. The actual CDW wave vector, as emphasized by Johannes and Mazin, is at the maximum of the real part of χ_0 , and for the purposes of the present paper it would probably be a better (or just another useful) quantity to consider.

Reviewer #4:

Remarks to the Author:

The authors have included transport data and provided a reasonable explanation of how the temperature was measured. Consequently, my previous concerns have been adequately addressed. I have no additional questions.

REVIEWER COMMENTS

Reviewer #2 (Remarks to the Author):

I thank the authors for the detailed response, which I find mostly convincing. Any further discussion would go into technical details that are probably not so important for the publication decision, so I tend to recommend the paper for publication in its current version.

We thank the reviewer for his time, questions and criticisms which have helped improve the manuscript. Below we have some final comments.

Below I just want to respond to the authors' replies to my comments.

1. Regarding the discussion of resolution effects, one has to understand that the assumption that the spectrum can be fitted by two symmetric peaks corresponding to Stokes and anti-Stokes processes with damping is an approximation. In reality, when considering both momentum and energy resolutions, the peak intensity will have contributions from an infinite number of phonons in the vicinity of the q vector. At the point where the dispersion relationship is nonlinear (which is the case at $E=0$), this can lead to an asymmetric shape of the peaks and the deviation of the fitted peak positions from the true peak positions. With this said, I admit that the one-dimensional type of analysis employed by the authors is considered standard in the field.

We agree with the referee. The momentum resolution can lead to such an effect. In our case, this could lead to slightly higher phonon energies than the true one (as exemplified by the data on SrTiO₃ in one of our previous communications). We believe that the effect is small. Yet, if considered, our data represent an upper boundary of the true energy. Thus, all softening effects, in particular at T^ , would only be underestimated by us – not overestimated. Technically, it is impossible to perform a realistic 3D momentum convolution as suggested by the referee since the dispersion at q_{CDW} is only known along the [100] direction.*

2. Regarding the calculation of the nesting vectors, the paper by Johannes and Mazin emphasizes the nesting occurs at wave vectors where the real part of the Lindhard function has its maxima. This is consistent with the criterion for the CDW instability, as described for example in S.-K. Chan and V. Heine, J. Phys. F: Met. Phys. 3 795 (1973). The quantity relevant for the formation of the CDW is χ_0 in the static limit. This has no contribution from the imaginary part which vanishes at $E=0$ as an odd function of energy. The confusion therefore seems to arise because different authors use the word "nesting" in different senses. If one means just overlapping Fermi surfaces (joint density of states), then the imaginary part of χ_0 must be considered, but this is unrelated to the CDW instability. The actual CDW wave vector, as emphasized by Johannes and Mazin, is at the maximum of the real part of χ_0 , and for the purposes of the present paper it would probably be a better (or just another useful) quantity to consider.

It seems that we misunderstood the previous comment of the referee interpreting his comments with respect to the imaginary part. Regarding the real part of the susceptibility and the work by Johannes & Mazin and Chan & Heine we agree with the referee. A peak in the imaginary part is not necessarily

connected to a phonon softening at the same wave vector. However, the soft phonon mode in TaSe2 features a clear peak in its momentum dependent linewidth as shown in Figure 6(b). The phonon linewidth is directly connected to the eJDOS and the nesting function. Thus a peak in the eJDOS can easily explain our observations for the linewidth and establishes a clear link to the Fermi surface geometry. Furthermore, our EPC calculations show that the position of the phonon softening and the peak in the eJDOS agree for 2H-TaSe2. Unfortunately, a calculation of the real part of the susceptibility is not included in our current code.

We included a statement in the supplementary information (red in the text below) regarding the relevance of the real part of the susceptibility and why, for 2H-TaSe₂, the imaginary part can be used to investigate the soft mode properties, too.

Supplemental Note 4: *Ab-initio* calculations

...We assessed the origin of the structural instability for ambient pressure lattice constants by calculating the electronic contribution to the linewidths of phonon modes,

$$\gamma_{EPC}^q = \pi\omega_{q\lambda} \sum_{\vec{k}} |g_{\vec{k}+\vec{q},\vec{k}}^{q\lambda}|^2 \delta(\epsilon_{\vec{k}} - \epsilon_F) \delta(\epsilon_{\vec{k}+\vec{q}} - \epsilon_F) \quad (1),$$

where $g_{\vec{k}+\vec{q},\vec{k}}^{q\lambda}$ is the EPC matrix elements for a phonon mode λ at a wave vector \vec{q} , $\omega_{q\lambda}$ is the mode's energy and $\sum_{\vec{k}} \delta(\epsilon_{\vec{k}} - \epsilon_F) \delta(\epsilon_{\vec{k}+\vec{q}} - \epsilon_F)$ gives the number of electronic states at the Fermi level connected by the phonon wave vector \vec{q} , i.e., the electronic joint density-of-states (eJDOS), which is practically equivalent to the nesting function¹ and the imaginary part of the static electronic susceptibility in the constant matrix-element approximation². **Generally, the wave vector of the phonon softening is connected to a maximum in the real part of the susceptibility³ which has been confirmed for 2H-TaSe₂.² However, the presence of a strongly momentum-dependent line width of the soft mode [see Fig. 6(b)] indicates that, for 2H-TaSe₂, the location of the phonon softening is also linked to a Fermi surface nesting geometry and, thus, can be investigated via the eJDOS.**

Reviewer #4 (Remarks to the Author):

The authors have included transport data and provided a reasonable explanation of how the temperature was measured. Consequently, my previous concerns have been adequately addressed. I have no additional questions.

We thank the referee for his time and effort reviewing our manuscript.

References:

- 1 Kasinathan, D., Kuneš, J., Lazicki, A., Rosner, H., Yoo, C. S., Scalettar, R. T. & Pickett, W. E. Superconductivity and Lattice Instability in Compressed Lithium from Fermi Surface Hot Spots. *Physical Review Letters* **96**, 047004 (2006).
- 2 Johannes, M. D. & Mazin, I. I. Fermi surface nesting and the origin of charge density waves in metals. *Physical Review B* **77**, 165135 (2008).
- 3 Chan, S. K. & Heine, V. Spin density wave and soft phonon mode from nesting Fermi surfaces. *Journal of Physics F: Metal Physics* **3**, 795 (1973).